# Generalized Venn and Venn-Abers Calibration with Applications in Conformal Prediction

**Lars van der Laan** [1]  **Ahmed Alaa** [2]

## Abstract

Ensuring model calibration is critical for reliable prediction, yet popular distribution-free methods such as histogram binning and isotonic regression offer only asymptotic guarantees. We introduce a unified framework for *Venn* and *Venn-Abers* calibration that extends Vovk's approach beyond binary classification to a broad class of prediction problems defined by generic loss functions. Our method transforms any perfectly *in-sample* calibrated predictor into a set-valued predictor that, in finite samples, outputs at least one *marginally* calibrated point prediction. These set predictions shrink asymptotically and converge to a conditionally calibrated prediction, capturing epistemic uncertainty. We further propose *Venn multicalibration*, a new approach for achieving finite-sample calibration across subpopulations. For quantile loss, our framework recovers group-conditional and multicalibrated conformal prediction as special cases and yields novel prediction intervals with quantile-conditional coverage.

## 1. Introduction

Calibration is essential for ensuring that machine learning models produce reliable predictions and enable robust decision-making across diverse applications. Model calibration aligns predicted probabilities with observed event frequencies and predicted quantiles with the specified proportion of outcomes. Recent work formalizes calibration as the alignment of predictions with elicitable properties defined through minimization of an expected loss, thereby generalizing traditional notions of mean and quantile calibration (Noarov and Roth, 2023). In safety-critical sectors

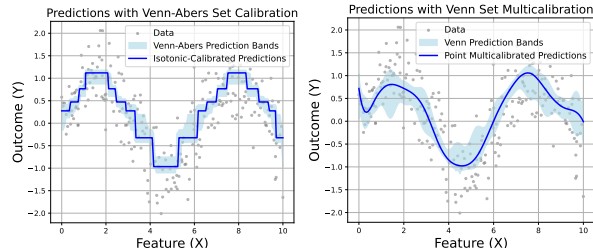

*Figure 1.* Prediction bands capturing epistemic uncertainty in the calibration of point predictions generated by our proposed methods with a squared error loss.

such as healthcare, it is crucial to ensure that model-driven decisions are reliable under minimal assumptions (Mandinach et al., 2006; Veale et al., 2018; Vazquez and Facelli, 2022; Gohar and Cheng, 2023). *Point calibrators*, which map a single model prediction to a single calibrated prediction (e.g., histogram binning and isotonic regression), can provide distribution-free calibration conditional on the calibration data. However, their guarantees are asymptotic, achieving zero calibration error only in the limit, and their performance may degrade in finite samples.

To address these limitations, *set calibrators* transform a single model prediction into a set of calibrated point predictions, explicitly capturing epistemic uncertainty in the calibration process. This class of methods includes Venn and Venn-Abers calibration (Vovk et al., 2003; Vovk and Petej, 2012; van der Laan and Alaa, 2024) for classification and regression, and Conformal Prediction (CP) for quantile regression and predictive inference (Vovk et al., 2005). Set calibrators provide prediction sets with finite-sample marginal calibration guarantees, while these sets generally converge asymptotically to conditionally calibrated point predictions. For example, Venn calibration produces a set of calibrated probabilities, and conformal prediction generates a set of calibrated quantiles (from which an interval can be derived), ensuring that at least one prediction in the set is perfectly calibrated marginally over draws of calibration data.

**Our Contributions.** We introduce a unified framework for Venn and Venn-Abers calibration, generalizing Vovk and Petej (2012) to a broad class of prediction tasks and loss functions. This framework extends point calibrators, e.g., histogram binning and isotonic regression, to produce pre-

---

[*]Equal contribution  [1]Department of Statistics, University of Washington [2]Computational Precision Health, UC Berkeley and UCSF. Correspondence to: Lars van der Laan <lvdlaan@uw.edu>.

*Proceedings of the 42nd International Conference on Machine Learning*, Vancouver, Canada. PMLR 267, 2025. Copyright 2025 by the author(s).

diction sets with finite-sample marginal and large-sample conditional calibration guarantees to capture epistemic uncertainty. We further propose *Venn multicalibration*, ensuring finite-sample calibration across subpopulations. For quantile regression, we show that Venn calibration corresponds to a novel CP procedure with quantile-conditional coverage, and that multicalibrated conformal prediction (Gibbs et al., 2023) is a special case of Venn multicalibration, unifying and extending existing calibration methods. Our approach enables the construction of set calibrators and set multicalibrators from point calibration algorithms for generic loss functions (Noarov and Roth, 2023).

## 2. Preliminaries for loss calibration

### 2.1. Notation

We consider the problem of predicting an outcome $Y \in \mathcal{Y}$ from a context $X \in \mathcal{X}$, where $Z := (X, Y)$ is drawn from an unknown distribution $P := P_X P_{Y|X}$, on which we impose no distributional assumptions. Let $f : \mathcal{X} \to \mathcal{Y}$ denote a predictive model trained to minimize a loss function $(f(x), z) \mapsto \ell(f(x), z)$ using a machine learning algorithm on training data. For example, $\ell(f(x), z)$ could be the squared error loss $\{y - f(x)\}^2$ or the $(1 - \alpha)$-quantile loss $\mathbb{1}\{y \geq f(x)\}\alpha(y - f(x)) + \mathbb{1}\{y < f(x)\}(1 - \alpha)(f(x) - y)$. We assume access to a calibration dataset $\mathcal{C}_n = \{(X_i, Y_i)\}_{i=1}^n$ of $n$ i.i.d. samples drawn from the same distribution $P$, independent from the training data. Throughout this work, we treat the model $f$ as fixed, implicitly conditioning on its training process.

### 2.2. Defining calibration for general losses

Machine learning models, such as neural networks and gradient-boosted trees, are powerful predictors but often produce biased point predictions due to model misspecification, distribution shifts, or limited data (Zadrozny and Elkan, 2001; Niculescu-Mizil and Caruana, 2005; Bella et al., 2010; Guo et al., 2017; Davis et al., 2017). To ensure reliable decision-making, we seek *calibrated* models (Roth, 2022; Silva Filho et al., 2023). Informally, calibration means that the predictions are optimal conditional on the prediction itself, so they cannot be improved by applying a transformation to reduce the loss. Formally, a model $f$ is perfectly $\ell$-calibrated if (Noarov and Roth, 2023; Whitehouse et al., 2024)

$$E_P[\ell(f(X), Z)] = \inf_\theta E_P[\ell(\theta(f(X)), Z)], \quad (1)$$

where the infimum is taken over all one-dimensional transformations $\theta : \mathbb{R} \to \mathbb{R}$ of $f$. Perfect calibration implies that $f(x) = \operatorname{argmin}_{c \in \mathbb{R}} E_P[\ell(c, Z) \mid f(X) = f(x)]$ for all $x \in \mathcal{X}$. We assume that the loss $\ell$ is smooth, such that $\ell$-calibration is equivalent to satisfying the first-order

conditions (Whitehouse et al., 2024):

$$E_P[\partial \ell(f(X), Z) \mid f(X) = f(x)] = 0 \text{ for all } x \in \mathcal{X}, \quad (2)$$

where $\partial \ell(f(x), y) := \frac{d}{d\eta}\ell(\eta, y)\mid_{\eta = f(x)}$ denotes the partial derivative of $\ell$ with respect to its first argument $f(x)$.

In regression, with $\ell$ taken as the squared error loss, $f$ is *perfectly calibrated* if $E_P[Y \mid f(X) = f(x)] = f(x)$ for all $x \in \mathcal{X}$ (Lichtenstein et al., 1977; Gupta et al., 2020), a property also known as self-consistency (Flury and Tarpey, 1996). This property addresses systematic over- or underestimation and ensures that $f(X)$ is a conditionally unbiased proxy for $Y$ in decision making. In binary classification, calibration ensures that the score $f(X)$ can be interpreted as a probability, making decision rules such as assigning label 1 when $f(X) > 0.5$ valid on average, since $P(Y = 1 \mid f(X) > 0.5) > 0.5$ (Silva Filho et al., 2023).

For a model $\widehat{f}$, which depends randomly on the calibration set $\mathcal{C}_n$, we define two types of calibration: marginal and conditional (Vovk, 2012). A random model $\widehat{f}$ is *conditionally* (perfectly) $\ell$-calibrated if it is calibrated conditional on the data $\mathcal{C}_n$: $E_P[\partial \ell(\widehat{f}(X), Z) \mid \widehat{f}(X), \mathcal{C}_n] = 0$ almost surely. The model $\widehat{f}$ is *marginally* $\ell$-calibrated if it is calibrated marginally over $\mathcal{C}_n$: $\mathbb{E}[\partial \ell(\widehat{f}(X), Z) \mid \widehat{f}(X)] = 0$, where $\mathbb{E}$ is taken over the randomness in both $(X, Y)$ and $\mathcal{C}_n$.

### 2.3. Post-hoc calibration via point calibrators

Models trained for predictive accuracy often require post-hoc calibration, typically using an independent calibration dataset or, in some cases, the training data (Gupta and Ramdas, 2021). Post-hoc methods treat prediction and calibration as distinct tasks, each optimized for a different yet complementary objective. Since perfect calibration is unattainable in finite samples, the goal of calibration is to obtain a model $\widehat{f}$ with minimal calibration error relative to a chosen metric, such as the conditional $\ell^2$ calibration error $\operatorname{Cal}_{\ell^2}(\widehat{f})$, defined as (Whitehouse et al., 2024):

$$\int \left\{ E_P[\partial \ell(\widehat{f}(X), Z) \mid \widehat{f}(X) = \widehat{f}(x), \mathcal{C}_n] \right\}^2 dP_X(x).$$

A *point calibrator*, following van der Laan et al. (2023) and Gupta et al. (2020), is a post-hoc procedure that learns a transformation $\theta_n : \mathbb{R} \to \mathbb{R}$ of a black-box model $f$ from $\mathcal{C}_n$ such that: (i) $\theta_n(f(X))$ is well-calibrated with low calibration error, and (ii) $\theta_n(f(X))$ remains comparably predictive to $f(X)$ in terms of the loss $\ell$. Common point calibrators include Platt scaling (Platt et al., 1999; Cox, 1958), histogram binning (Zadrozny and Elkan, 2001), and isotonic calibration (Zadrozny and Elkan, 2002).

A calibrated predictor can be constructed from $\mathcal{C}_n$ by learning the calibrator $\theta_n$ via minimizing the empirical risk

$\sum_{i=1}^{n} \ell(\theta(f(X_i)), Z_i)$. For regression, this involves regressing outcomes $\{Y_i\}_{i=1}^{n}$ on predictions $\{f(X_i)\}_{i=1}^{n}$ (Mincer and Zarnowitz, 1969). If $\theta_n$ accurately estimates the conditional calibration function $\min_\theta E_P[\ell(\theta(f(X)), Z) \mid \mathcal{C}_n]$, the predictor is well-calibrated by the tower property. This approach is not distribution-free and typically requires correct model specification. Methods like kernel smoothing and Platt scaling impose smoothness or parametric assumptions on the calibration function (Jiang et al., 2011).

### 2.4. Distribution-free calibration via histogram binning

A simple class of distribution-free calibration methods is histogram binning, such as uniform mass (quantile) binning (Gupta et al., 2020; Gupta and Ramdas, 2021). Here, the prediction space $f(\mathcal{X})$ is partitioned into $K$ bins $\{B_k\}_{k=1}^{K}$ in an outcome-agnostic manner, often by taking quantiles of the predictions $\{f(X_i)\}_{i=1}^{n}$ in $\mathcal{C}_n$. The binning calibrator $\theta_n$ is a step function obtained via histogram regression:

$$\theta_n(t) := \min_{c \in \mathbb{R}} \sum_{i=1}^{n} \mathbb{1}\{f(X_i) \in B_{k(t)}\}\ell(c, Z_i),$$

where $k(t)$ indexes the bin containing $t \in f(\mathcal{X})$. For the squared error loss, $\theta_n(t)$ is the empirical mean of the outcomes $\{Y_i : i \in [n], f(X_i) \in B_{k(t)}\}$ within the bin $B_{k(t)}$. The histogram regression property ensures that the resulting calibrated model $f_n^* := \theta_n \circ f$ is *in-sample $\ell$–calibrated* (van der Laan et al., 2024b), meaning that, for each $x \in \mathcal{X}$,

$$\sum_{i=1}^{n} \mathbb{1}\{f_n^*(X_i) = f_n^*(x)\}\partial\ell(f_n^*(x), Z_i) = 0, \qquad (3)$$

implying that its empirical risk cannot be improved by any transformation of its predictions.

in-sample calibration does not guarantee good out-of-sample calibration or predictive performance. With a maximal partition $K = n$, perfect in-sample calibration leads to overfitting, resulting in poor out-of-sample performance. Conversely, with a minimal partition $K = 1$, the model is well-calibrated but poorly predictive, yielding a constant predictor $\min_{c \in \mathbb{R}} \sum_{i=1}^{n} \ell(c, Z_i)$. This illustrates a trade-off: too few bins reduce predictive power, while too many increase variance and degrade calibration. Histogram binning asymptotically provides conditionally calibrated predictions, with the conditional $\ell^2$ calibration error satisfying $\mathrm{Cal}_{\ell^2}(f_n^*) = O_p\left(\frac{K \log(n/K)}{n}\right)$ (Whitehouse et al., 2024). Thus, tuning of $K$, for example via cross-validation, is crucial to balance calibration and predictiveness.

## 3. Generalized Venn calibration framework

### 3.1. Venn calibration for general losses

Suppose we are interested in obtaining an $\ell$-calibrated prediction $f(X_{n+1})$ for an unseen outcome $Y_{n+1}$ from a new context $X_{n+1}$, where $(X_{n+1}, Y_{n+1})$ is drawn from $P$ and independent of the calibration data $\mathcal{C}_n$. Let $\mathcal{A}_\ell$ be any point calibration algorithm that takes a model $f$ and calibration data $\mathcal{C}_n$ and outputs a refined model $f_n^* := \mathcal{A}_\ell(f, \mathcal{C}_n)$ that is in-sample $\ell$–calibrated in the sense of (3). For example, $\mathcal{A}_\ell(f, \mathcal{C}_n)$ could be defined as $\theta_n \circ f$, where $\theta_n$ is learned using an outcome-agnostic histogram binning method, such as uniform mass binning, or an outcome-adaptive method, such as a regression tree or isotonic regression. We assume that the algorithm $\mathcal{A}_\ell$ processes input data exchangeably, ensuring the calibrated predictor is invariant to permutations of the calibration data.

Distribution-free binning-based point calibrators, like histogram binning and isotonic regression, achieve perfect in-sample calibration on calibration data but may exhibit poor conditional calibration in finite samples, attaining population-level calibration only asymptotically. To address this, Vovk et al. (2003) proposed Venn calibration for binary classification, which transforms point calibrators into set calibrators that ensure finite-sample marginal calibration while retaining conditional calibration asymptotically.

In this section, we extend Venn calibration to general prediction tasks defined by loss functions. We demonstrate that Venn calibration can be applied to any point calibrator that achieves perfect $\ell$-*in-sample* calibration to ensure finite-sample *marginal* calibration. Unlike traditional point calibrators, which produce a single calibrated prediction for each context, Venn calibration generates a prediction set that is guaranteed to contain a marginally perfectly $\ell$-calibrated prediction. This set reflects epistemic uncertainty by covering a range of possible calibrated predictions, each of which remains asymptotically conditionally $\ell$-calibrated. A prominent example is Venn-Abers calibration, which uses isotonic regression as the underlying point calibrator (Vovk and Petej, 2012; Toccaceli, 2021; van der Laan and Alaa, 2024).

Our generalized Venn calibration procedure is detailed in Alg. 1. A distinctive aspect of Venn calibration is that it adapts the model $f$ specifically for the given context $X_{n+1}$, unlike point calibrators, which produce a single calibrated model intended to be (asymptotically) valid across all contexts. For a given context $X_{n+1}$, the algorithm iteratively considers imputed outcomes $y \in \mathcal{Y}$ for $Y_{n+1}$ and applies the calibrator $\mathcal{A}_\ell$ to the augmented dataset $\mathcal{C}_n \cup \{(X_{n+1}, y)\}$. This process yields a set of point predictions:

$$f_{n, X_{n+1}}(X_{n+1}) := \{f_n^{(X_{n+1}, y)}(X_{n+1}) : y \in \mathcal{Y}\}.$$

When the outcome space $\mathcal{Y}$ is continuous, Alg. 1 may be computationally infeasible to execute exactly and can instead be approximated by discretizing $\mathcal{Y}$. Nonetheless, the range of the prediction set $f_{n,x}(x)$ can often be computed by iterating over the extreme points $\{y_{\min}, y_{\max}\}$ of $\mathcal{Y}$.

---

**Algorithm 1** Venn loss calibration

---

**Require:** Calibration data $\mathcal{C}_n = \{(X_i, Y_i)\}_{i=1}^n$, model $f$, context $x \in \mathcal{X}$, loss calibrator $\mathcal{A}_\ell$.

1: **for** each $y \in \mathcal{Y}$ **do**
2:     augment dataset: $\mathcal{C}_n^{(x,y)} := \mathcal{C}_n \cup \{(x,y)\}$;
3:     calibrate model: $f_n^{(x,y)} := \mathcal{A}_\ell(f, \mathcal{C}_n^{(x,y)})$;
4: **end for**
5: set $f_{n,x}(x) := \{f_n^{(x,y)}(x) : y \in \mathcal{Y}\}$;
**Ensure:** prediction set $f_{n,x}(x)$.

---

To establish the validity of Venn calibration, we impose the following conditions, which ensure that the data are exchangeable — a common assumption in conformal prediction (Vovk et al., 2005), particularly satisfied when the data are i.i.d — and that the derivative of the loss has a finite second moment.

**C1)** *Exchangeability:* $\{(X_i, Y_i)\}_{i=1}^{n+1}$ are exchangeable.

**C2)** *Finite variance:* $\mathbb{E}[\{\partial\ell(f_{n+1}^*(X_{n+1}), Z_{n+1})\}^2] < \infty$.

**C3)** *Perfect in-sample calibration:* $\sum_{i=1}^{n+1} \mathbb{1}\{f_{n+1}^*(X_i) = f_{n+1}^*(x)\}\partial\ell(f_{n+1}^*(X_i), Z_i) = 0$ almost surely for each $x \in \mathcal{X}$.

The finite-sample validity of the Venn calibration procedure can be established through an *oracle* procedure that assumes knowledge of the unseen outcome $Y_{n+1}$. In this oracle procedure, a perfectly in-sample $\ell$-calibrated prediction $f_{n+1}^*(X_{n+1}) := \mathcal{A}_\ell(f, \mathcal{C}_{n+1}^*)(X_{n+1})$ is obtained by calibrating $f$ using $\mathcal{A}_\ell$ on the oracle-augmented calibration set $\mathcal{C}_{n+1}^* := \mathcal{C}_n \cup \{(X_{n+1}, Y_{n+1})\}$. By leveraging exchangeability, the in-sample calibration of the oracle prediction $f_{n+1}^*(X_{n+1})$ ensures *marginal* perfect $\ell$-calibration, such that $\mathbb{E}[\partial\ell(f_{n+1}^*(X_{n+1}), Z_{n+1}) \mid f_{n+1}^*(X_{n+1})] = 0$ almost surely. Since the oracle prediction is, by construction, contained in the Venn prediction set $f_{n,X_{n+1}}(X_{n+1})$, we conclude the following theorem.

**Theorem 3.1** (Marginal calibration of Venn prediction). *Under C1-C3, the Venn prediction set $f_{n,X_{n+1}}(X_{n+1})$ contains the marginally perfectly $\ell$-calibrated prediction, $f_{n+1}^*(X_{n+1}) = f_n^{(X_{n+1}, Y_{n+1})}$, which satisfies*

$$\mathbb{E}\left[\{\mathbb{E}[\partial\ell(f_{n+1}^*(X_{n+1}), Z_{n+1}) \mid f_{n+1}^*(X_{n+1})]\}^2\right] = 0.$$

In order to satisfy C3, Algorithm 1 should be applied with a binning-based calibrator $\mathcal{A}_\ell$ that achieves perfect in-sample

calibration, such as uniform mass binning, a regression tree, or isotonic regression. Importantly, this condition does not impose restrictions on how the bins are selected, allowing pre-specified and data-adaptive binning schemes. While the choice of binning calibrator does not affect the marginal perfect calibration guarantee in Theorem 3.1, it influences both the width of the Venn prediction set and the conditional calibration of the point predictions. In general, using a point calibrator that ensures conditionally well-calibrated predictions, such as histogram binning with appropriately tuned bins or isotonic calibration, is recommended.

For example, in the regression setting with squared error loss, Theorem 3.1 ensures that the set $f_{n, X_{n+1}}(X_{n+1})$ contains $f_{n+1}^*(X_{n+1}) = \mathbb{E}[Y_{n+1} \mid f_{n+1}^*(X_{n+1})]$. However, using histogram binning with one observation per bin results in $f_{n,X_{n+1}}(X_{n+1}) = \mathcal{Y}$, an uninformative set that poorly reflects conditional calibration despite including $f_{n+1}^*(X_{n+1}) = Y_{n+1}$. In contrast, using a single bin produces a set containing the marginally perfectly calibrated prediction $f_{n+1}^*(X_{n+1}) = \frac{1}{n+1}\sum_{i=1}^{n+1} Y_i$, where each prediction is close to the sample mean $\frac{1}{n}\sum_{i=1}^n Y_i$, ensuring conditional calibration but resulting in poor predictiveness.

Suppose that $\mathcal{A}_\ell(f, \mathcal{C}_n \cup \{x, y\})$ outputs a transformed model $\theta_n^{(x,y)} \circ f$, where $\theta_n^{(x,y)}$ is learned using an outcome-agnostic or outcome-adaptive binning calibrator, such as quantile binning or a regression tree. The following theorem shows that each calibrated model $f_n^{(x,y)} = \theta_n^{(x,y)} \circ f$, used to construct the Venn prediction set, is asymptotically conditionally calibrated, provided the calibration data are i.i.d. and the number of bins in the calibrator $\theta_n^{(x,y)}$ does not grow too quickly.

**C4)** *Independence:* $\{(X_i, Y_i)\}_{i=1}^{n+1}$ are i.i.d.

**C5)** *Boundedness:* $\operatorname{ess\,sup}_{z'=(x',y')} |\partial\ell(\theta_n^{(x,y)}(f(x')), z')|$ and $\operatorname{ess\,sup}_{x'} |\theta_n^{(x,y)}(x')|$ are bounded by a constant $M < \infty$.

**C6)** *Lipschitz derivative:* There exists $L < \infty$ such that $|\partial\ell(\eta_1, z) - \partial\ell(\eta_2, z)| \le L|\eta_1 - \eta_2|$ for all $z, \eta_1, \eta_2$.

**C7)** *Finite number of bins:* $\theta_n^{(x,y)}$ is piecewise constant taking at most $k(n) < \infty$ values.

**Theorem 3.2** (Conditional calibration of Venn calibration). *Under C4-C7, we have* $\mathrm{Cal}_{\ell^2}(f_n^{(x,y)}) = O_p\left(\frac{k(n)\log(n/k(n))}{n}\right)$.

For stable point calibrators, as the calibration set size $n$ increases, the Venn prediction set narrows and converges to a single perfectly $\ell$-calibrated prediction (Vovk and Petej, 2012). In large-sample settings, where standard point calibrators perform reliably, the Venn prediction set becomes

narrow, closely resembling a point prediction. In contrast, in small-sample settings, where overfitting can undermine the reliability of point calibrators such as histogram binning and isotonic calibration, the Venn prediction set widens, reflecting increased uncertainty about the true calibrated prediction (Johansson et al., 2023). Consequently, Venn calibration improves the robustness of the point calibration procedure $\mathcal{A}_\ell$ by explicitly representing uncertainty through a set of possible calibrated predictions.

### 3.2. Isotonic and Venn-Abers calibration

Venn calibration can be applied with any loss calibrator $\mathcal{A}_\ell$ that provides in-sample calibrated predictions. While histogram binning requires pre-specifying the number of bins, isotonic calibration (Zadrozny and Elkan, 2002; Niculescu-Mizil and Caruana, 2005) addresses this limitation by adaptively determining bins through isotonic regression, a non-parametric method for estimating monotone functions (Barlow and Brunk, 1972). Instead of fixing $K$ in advance, isotonic calibration selects bins by minimizing an empirical MSE criterion, ensuring the calibrated predictor is a non-decreasing monotone transformation of the original predictor. Isotonic calibration allows the number of bins to grow with sample size, ensuring good calibration while preserving predictive performance. van der Laan et al. (2023) show that, in the context of treatment effect estimation, the conditional $\ell^2$ calibration error of isotonic calibration for i.i.d. data asymptotically satisfies $\mathrm{Cal}_{\ell^2}(f_n^*) = O_p(n^{-2/3})$.

---

**Algorithm 2** Venn-Abers loss calibration

**Require:** Calibration data $\mathcal{C}_n = \{(X_i, Y_i)\}_{i=1}^n$, model $f$, loss $\ell$, context $x \in \mathcal{X}$.

1: **for** each $y \in \mathcal{Y}$ **do**
2:     augment dataset: $\mathcal{C}_n^{(x,y)} := \mathcal{C}_n \cup \{(X_{n+1}, Y_{n+1}) := (x, y)\}$;
3:     calibrate model using generalized isotonic regression:

$$\theta_n^{(x,y)} := \mathrm{argmin}_{\theta \in \Theta_{\mathrm{iso}}} \sum_{i \in \mathcal{C}_n^{(x,y)}} \ell(\theta(f(X_i)), Z_i).$$
$$f_n^{(x,y)} := \theta_n^{(x,y)} \circ f.$$

4: **end for**
5: set $f_{n,x}(x) := \{f_n^{(x,y)}(x) : y \in \mathcal{Y}\}$;

**Ensure:** prediction set $f_{n,x}(x)$.

---

In this section, we propose Venn-Abers calibration for general loss functions, a special instance of Venn calibration that employs isotonic regression as the underlying point calibrator, thereby generalizing the original procedure for classification and regression (Vovk and Petej, 2012; van der Laan and Alaa, 2024). Our generalized Venn-Abers calibration procedure is outlined in Alg. 2. Isotonic regression is a stable algorithm, meaning small changes in the training set do not significantly affect the solution, ensuring that the Venn-Abers prediction set converges to a point prediction as the sample size grows (Caponnetto and Rakhlin, 2006; Bous-

quet and Elisseeff, 2000). Consequently, the Venn-Abers prediction set inherits the marginal calibration guarantee of Venn calibration, while each point prediction in the set is conditionally calibrated in large samples.

Let $f_{n,X_{n+1}}(X_{n+1})$ denote the Venn-Abers prediction set obtained by applying Alg. 2 with $x = X_{n+1}$. The following theorem follows directly from Theorem 3.1.

**Theorem 3.3** (Marginal calibration of Venn-Abers). *Under C1 and C2, the Venn prediction set $f_{n,X_{n+1}}(X_{n+1})$ contains the marginally perfectly $\ell$-calibrated prediction, $f_{n+1}^*(X_{n+1}) := f_n^{(X_{n+1}, Y_{n+1})}$, which satisfies*

$$\mathbb{E}\left[\left\{\mathbb{E}[\partial \ell(f_{n+1}^*(X_{n+1}), Z_{n+1}) \mid f_{n+1}^*(X_{n+1})]\right\}^2\right] = 0.$$

The following theorem establishes that each isotonic calibrated model $f_n^{(x,y)}$ used to construct the Venn-Abers prediction set is asymptotically conditionally calibrated.

**C8)** *Best predictor of gradient has finite variation:* There exists an $B < \infty$ such that $t \mapsto E_P[\partial \ell(f_n^{(x,y)}(X), Y) \mid f(X) = t, \mathcal{C}_n]$ has total variation norm that is almost surely bounded by $B$.

**Theorem 3.4** (Conditional calibration of Venn-Abers). *Under C4-C6, and C8, we have $\mathrm{Cal}_{\ell^2}(f_n^{(x,y)}) = O_p(n^{-2/3})$.*

This theorem generalizes the distribution-free conditional calibration guarantees for isotonic calibration of van der Laan et al. (2023) and van der Laan et al. (2024a) for regression and inverse probabilities to general losses.

**Computational considerations.** As discussed in van der Laan and Alaa (2024), the main computational cost of Alg. 2 lies in the isotonic calibration step for each $y \in \mathcal{Y}$. Isotonic regression (Barlow and Brunk, 1972) can be efficiently computed using `xgboost` (Chen and Guestrin, 2016) with monotonicity constraints. Similar to Full CP (Vovk et al., 2005), Alg. 2 may be infeasible for non-discrete outcomes, but it can be approximated by iterating over a finite subset of $\mathcal{Y}$ with linear interpolation for $f_n^{(x,y)}(x)$. Like Full and multicalibrated CP (Gibbs et al., 2023), this algorithm must be applied separately for each context $x \in \mathcal{X}$. Since the algorithms depend on $x \in \mathcal{X}$ only through its prediction $f(x)$, we can approximate the outputs for all $x \in \mathcal{X}$ by running each algorithm on a finite set of $x \in \mathcal{X}$ corresponding to a finite grid over the one-dimensional output space $f(\mathcal{X}) = \{f(x) : x \in \mathcal{X}\} \subset \mathbb{R}$. Moreover, both algorithms are fully parallelizable across both the input context $x \in \mathcal{X}$ and the imputed outcome $y \in \mathcal{Y}$. In our implementation, we use nearest neighbor interpolation in the prediction space to impute outputs for each $x \in \mathcal{X}$. In our experiments with sample sizes ranging from $n = 5000$ to $40000$, quantile binning of both $f(\mathcal{X})$ and $\mathcal{Y}$ into 200 equal-frequency bins

enables execution of Algorithm 2 with squared error and quantile loss across all contexts in minutes, with negligible approximation error.

### 3.3. Venn multicalibration for finite-dimensional classes

Standard calibration ensures that predicted outcomes cannot be improved by any transformation of the predictions, making them optimal on average for contexts with the same predicted value. However, this aggregate guarantee can mask systematic errors within subgroups. Multicalibration extends standard calibration by enforcing calibration within specified subpopulations, ensuring fairness and reliability across groups (Jung et al., 2008; Roth, 2022; Noarov and Roth, 2023; Deng et al., 2023; Haghtalab et al., 2023). In this section, we introduce Venn multicalibration, a generalization of Venn calibration that provides calibration guarantees across multiple subpopulations. This approach produces prediction sets that contain a perfectly multicalibrated prediction in finite samples. Our work enables the extension of existing methods for pointwise multicalibration with generic loss functions to set-valued calibration (Noarov and Roth, 2023; Deng et al., 2023; Haghtalab et al., 2023).

We say a model $\widehat{f}$ is *marginally perfectly $\ell$-multicalibrated* with respect to a function class $\mathcal{G}$ if the following holds:

$$\mathbb{E}\left[\frac{\partial}{\partial t}\ell((\widehat{f} + tg)(X_{n+1}), Z_{n+1})\,|_{t=0}\right] = 0 \text{ for all } g \in \mathcal{G},$$
(4)

where the expectation is taken over $(X_{n+1}, Y_{n+1})$ as well as randomness of $\widehat{f}$. Assuming the order of integration and differentiation can be exchanged, this condition implies that

$$\mathbb{E}\left[\ell(\widehat{f}(X_{n+1}), Z_{n+1})\right] = \min_{g \in \mathcal{G}} \mathbb{E}\left[\ell((\widehat{f} + g)(X_{n+1}), Y_{n+1})\right].$$

In other words, the loss $\ell(\widehat{f}(X_{n+1}), Z_{n+1})$ incurred for the new data point cannot be improved in expectation by adjusting the calibrated model $\widehat{f}$ using functions from $\mathcal{G}$.

Multicalibration for classification and regression requires that (Hébert-Johnson et al., 2018; Kim et al., 2019):

$$\mathbb{E}\left[g(X_{n+1})\{Y_{n+1} - \widehat{f}(X_{n+1})\}\right] = 0 \text{ for all } g \in \mathcal{G}. \quad (5)$$

The function $g \in \mathcal{G}$ is sometimes viewed to as a representation of covariate shift. When $g$ is nonnegative, it follows that $\mathbb{E}_g[\widehat{f}(X_{n+1})] = \mathbb{E}_g[Y_{n+1}]$, where the weighted expectation is defined as $\mathbb{E}_g[\widehat{f}(X_{n+1})] = \mathbb{E}\left[\frac{g(X_{n+1})}{\mathbb{E}[g(X_{n+1})]}\widehat{f}(X_{n+1})\right]$. For example, when $\mathcal{G} = \{x \mapsto \mathbb{1}(x \in B) : B \in \mathcal{B}\}$ consists of all set indicators for (possibly intersecting) subgroups in $\mathcal{B}$, multicalibration implies that the model $\widehat{f}(X_{n+1})$ is calibrated for $Y_{n+1}$ within each subgroup, meaning that $\mathbb{E}[Y_{n+1} \mid X_{n+1} \in B] = \mathbb{E}[\widehat{f}(X_{n+1}) \mid X_{n+1} \in B]$.

---

**Algorithm 3** Venn loss multicalibration

**Require:** Calibration data $\mathcal{C}_n = \{(X_i, Y_i)\}_{i=1}^n$, model $f$, loss $\ell$, context $x \in \mathcal{X}$, function class $\mathcal{F}$.

1: **for** each $y \in \mathcal{Y}$ **do**
2:     augment dataset: $\mathcal{C}_n^{(x,y)} := \mathcal{C}_n \cup \{(X_{n+1}, Y_{n+1}) := (x, y)\}$;
3:     multicalibrate model using offset loss minimization:

       $g_n^{(x,y)} := \operatorname{argmin}_{g \in \mathcal{G}} \sum_{i \in \mathcal{C}_n^{(x,y)}} \ell(f(X_i) + g(X_i), Z_i)$.
       $f_n^{(x,y)} := f + g_n^{(x,y)}$.

4: **end for**
5: set $f_{n,x}(x) := \{f_n^{(x,y)}(x) : y \in \mathcal{Y}\}$;
**Ensure:** prediction set $f_{n,x}(x)$.

---

For a finite-dimensional function class $\mathcal{G}$, we propose Venn multicalibration for a generic loss function $\ell$ in Algorithm 3. For mean multicalibration with squared error loss, this algorithm can be computed efficiently using the Sherman–Morrison formula to update linear regression solutions with new data points (Shermen and Morrison, 1949; Yang et al., 2023). This formula has previously been used for efficient computation of leave-one-out (e.g., Jackknife) predictions by applying rank-one updates to the inverse Gram matrix, thereby enabling fast updates to the least-squares solution without retraining on each leave-one-out subset. Under monotonicity of $y \mapsto f_n^{(x,y)}$, the range of the Venn prediction set $f_{n,x}(x)$ can be computed by iterating over the extreme points $\{y_{\min}, y_{\max}\}$ of $\mathcal{Y}$.

The following theorem establishes that the prediction set $f_{n,X_{n+1}}(X_{n+1})$ in Alg. 3 contains the $\ell$-multicalibrated prediction $f_{n+1}^*(X_{n+1})$, where $f_{n+1}^* := f_n^{(X_{n+1}, Y_{n+1})}$.

**C9)** *In-sample multicalibration:* $\sum_{i=1}^{n+1} \frac{\partial}{\partial t}\ell((f_{n+1}^* + tg)(X_i), Z_i)\,|_{t=0} = 0$ almost surely for each $g \in \mathcal{G}$.

**Theorem 3.5** (Perfect calibration of Venn multicalibration)**.** *Under C1 and C9, the Venn prediction set $f_{n,X_{n+1}}(X_{n+1})$ contains the marginally perfectly calibrated prediction $f_{n+1}^*(X_{n+1}) = f_n^{(X_{n+1}, Y_{n+1})}(X_{n+1})$, which satisfies*

$$\mathbb{E}\left[\frac{\partial}{\partial t}\ell((f_{n+1}^* + tg)(X_{n+1}), Z_{n+1})\,|_{t=0}\right] = 0, \forall g \in \mathcal{G}.$$

In the special case where $\ell$ is the squared error loss, the next corollary shows that Venn prediction sets contain a perfectly multicalibrated regression prediction, as defined in (5).

**Corollary 3.6** (Regression Venn Multicalibration)**.** *Suppose that $\ell(f(x), z))$ is $\{y - f(x)\}^2$. Under C1 and C2, the Venn prediction set $f_{n,X_{n+1}}(X_{n+1})$ contains the perfectly multicalibrated prediction, denoted as $f_{n+1}^*(X_{n+1}) := f_n^{(X_{n+1}, Y_{n+1})}(X_{n+1})$, which satisfies $\mathbb{E}\left[g(X_{n+1})\{Y_{n+1} - f_{n+1}^*(X_{n+1})\}\right] = 0$ for all $g \in \mathcal{G}$.*

# 4. Applications to conformal prediction

## 4.1. Conformal prediction via Venn quantile calibration

Conformal prediction (CP) (Vovk et al., 2005) is a flexible approach for predictive inference that can be applied post-hoc to any black-box model. It constructs prediction intervals $\widehat{C}_n(X_{n+1})$ that are guaranteed to *cover* the true outcome $Y_{n+1}$ with probability $1 - \alpha$. The standard CP method ensures prediction intervals satisfy the marginal coverage guarantee: $\mathbb{P}(Y_{n+1} \in \widehat{C}_n(X_{n+1})) \geq 1 - \alpha$, where the probability $\mathbb{P}$ accounts for the randomness in $\mathcal{C}_n$ and $(X_{n+1}, Y_{n+1})$. In this section, we show how our generalized Venn calibration framework, when combined with the quantile loss, enables the construction of perfectly calibrated quantile predictions and CP intervals in finite samples.

Let $\{S_i\}_{i=1}^{n+1}$ be conformity scores, where $S_i = \mathcal{S}(X_i, Y_i)$ for some scoring function $\mathcal{S}$. For example, we could set $\mathcal{S}$ as the absolute residual scoring function $z \mapsto |y - \mu(x)|$, where $\mu$ predicts $y$ from $x$. Conformity scores quantify how well a predicted outcome aligns with the true outcome. Let $f : \mathcal{X} \to \mathbb{R}$ be a model trained to predict the $(1-\alpha)$ quantile of the conformity scores. Given $f$, we can define a conformal interval for $Y_{n+1}$ as $\{y \in \mathcal{Y} : \mathcal{S}((X_{n+1}, y)) \leq f(X_{n+1})\}$. However, this interval does not provide distribution-free coverage guarantees due to potential miscalibration of $f$. To ensure finite-sample coverage, we propose calibrating the predictor using quantile Venn and Venn-Abers calibration.

For a quantile level $\alpha \in (0, 1)$, we denote the quantile loss $\ell_\alpha(q, y)$ by $\mathbb{1}(y \geq q) \cdot \alpha(y - q) + \mathbb{1}(y < q) \cdot (1 - \alpha)(q - y)$. Let $f_{n,X_{n+1}}(X_{n+1}) = \{f_n^{(X_{n+1}, y)}(X_{n+1}) : y \in \mathcal{Y}\}$ be the Venn quantile prediction set of $S_{n+1}$ obtained by applying Algorithm 2 with the conformal quantile loss $\ell_{\mathcal{S},\alpha} : (f, z) \mapsto \ell_\alpha(f, \mathcal{S}(z))$, and let $f_{n+1}^*(X_{n+1})$ be the perfectly calibrated prediction in this set, where $f_{n+1}^* := f_n^{(X_{n+1}, Y_{n+1})}$ equals $\mathcal{A}_\ell(f, \{(X_i, Y_i)\}_{i=1}^{n+1})$. The Venn prediction set $f_{n,X_{n+1}}(X_{n+1})$ induces the Venn CP interval:

$$\widehat{C}_n(X_{n+1}) := \{y \in \mathcal{Y} : \mathcal{S}((X_{n+1}, y)) \leq f_n^{(X_{n+1}, y)}(X_{n+1})\}$$

**C10)** *In-sample calibration:* $\sum_{i=1}^{n+1} \ell_{\mathcal{S},\alpha}(Z_i, f_{n+1}^*) = \min_\theta \sum_{i=1}^{n+1} \ell_{\mathcal{S},\alpha}(Z_i, \theta \circ f_{n+1}^*)$.

**Theorem 4.1** (Calibration of Venn Quantile Prediction). *Assume C1, C10, and that $S_i \neq f_{n+1}^*(X_i)$ almost surely for each $i \in [n + 1]$. Then, the Venn prediction set $f_{n,X_{n+1}}(X_{n+1})$ contains the marginally perfectly calibrated prediction $f_{n+1}^*(X_{n+1}) = f_n^{(X_{n+1}, Y_{n+1})}(X_{n+1})$, where $\mathbb{P}(S_{n+1} \leq f_{n+1}^*(X_{n+1}) \mid f_{n+1}^*(X_{n+1})) = 1 - \alpha$.*

Theorem 4.1 implies that the Venn CP interval $\widehat{C}_n(X_{n+1})$ constructed using Venn quantile calibration is perfectly calibrated in that $\mathbb{P}(Y_{n+1} \in \widehat{C}_n(X_{n+1}) \mid f_{n+1}^*(X_{n+1})) =$

$1 - \alpha$. This result follows directly from the perfect quantile calibration of $f_{n+1}^*(X_{n+1})$ because, by definition,

$$\mathbb{P}(Y_{n+1} \in \widehat{C}_n(X_{n+1}) \mid f_{n+1}^*(X_{n+1}))$$
$$= \mathbb{P}(\mathcal{S}((X_{n+1}, Y_{n+1})) \leq f_n^{(X_{n+1}, Y_{n+1})} \mid f_{n+1}^*(X_{n+1}))$$
$$= \mathbb{P}(S_{n+1} \leq f_{n+1}^*(X_{n+1}) \mid f_{n+1}^*(X_{n+1}))$$
$$= 1 - \alpha.$$

As a consequence, the CP interval satisfies a form of *threshold calibration* (Jung et al., 2022), meaning its coverage is valid conditional on the quantile $f_{n+1}^*(X_{n+1})$ used to define the interval. The law of total expectation implies that $\widehat{C}_n(X_{n+1})$ also satisfies the marginal calibration condition $\mathbb{P}(S_{n+1} \leq f_{n+1}^*(X_{n+1})) = 1 - \alpha$. We note that, without assuming $S_i \neq f_{n+1}^*(X_i)$ almost surely for each $i \in [n + 1]$, we can still establish the lower bound $\mathbb{P}(Y_{n+1} \in \widehat{C}_n(X_{n+1}) \mid f_{n+1}^*(X_{n+1})) \geq 1 - \alpha$ using arguments from Gibbs et al. (2023), though we do not pursue this here for simplicity.

## 4.2. Conformal prediction as Venn multicalibration

In this section, we show that conformal prediction is a special case of Venn multicalibration with the quantile loss.

Suppose Alg. 3 is applied with the conformal quantile loss $(f(x), z) \mapsto \ell_\alpha(f(x), \mathcal{S}(z))$, model $f$, and $x := X_{n+1}$, and let $f_{n,X_{n+1}}(X_{n+1})$ be the corresponding Venn set prediction. As in Section 4.1, we define the multicalibrated Venn CP interval as $\widehat{C}_n(X_{n+1}) := \{y \in \mathcal{Y} : \mathcal{S}((X_{n+1}, y)) \leq f_n^{(X_{n+1}, y)}(X_{n+1})\}$. This multicalibrated CP interval is identical to the interval proposed in the conditional CP framework of Gibbs et al. (2023).

The following theorem shows that Venn multicalibration outputs a prediction set containing a marginally multicalibrated quantile prediction (Deng et al., 2023), ensuring that the CP interval is multicalibrated in the sense of Gibbs et al. (2023).

**Theorem 4.2** (Quantile Multicalibration). *Assume C1 holds and $S_i \neq f_{n+1}^*(X_i)$ almost surely for all $i \in [n + 1]$. Then, the Venn prediction set $f_{n,X_{n+1}}(X_{n+1})$ contains the perfectly multicalibrated prediction $f_{n+1}^*(X_{n+1}) := f_n^{(X_{n+1}, Y_{n+1})}(X_{n+1})$, which satisfies $\mathbb{E}\left[g(X_{n+1})\{(1 - \alpha) - \mathbb{P}(S_{n+1} \leq f_{n+1}^*(X_{n+1}) \mid X_{n+1})\}\right] = 0$ for all $g \in \mathcal{G}$.*

By definition, Theorem 4.2 implies the multicalibrated coverage of the conformal interval: for all $g \in \mathcal{G}$,

$$\mathbb{E}\left[g(X_{n+1})\{(1 - \alpha) - \mathbb{P}(Y_{n+1} \in \widehat{C}_n(X_{n+1}) \mid X_{n+1})\}\right] = 0,$$

which agrees with Theorem 2 of Gibbs et al. (2023).

As a consequence, the multicalibrated CP framework for finite-dimensional covariate shifts proposed in Section 2.2

of Gibbs et al. (2023) can be interpreted as a special case of Venn multicalibration. Similarly, the standard marginal CP approach (Vovk et al., 2005; Lei et al., 2018) and Mondrian (or group-conditional) CP (Vovk et al., 2005; Romano et al., 2020) are special cases of this algorithm, with $\mathcal{G}$ consisting of constant functions and subgroup indicators, respectively.

# 5. Numerical experiments

The utility of Venn and Venn-Abers calibration for classification and regression, as well as Venn multicalibration with the quantile loss in the context of conformal prediction (CP), has been demonstrated through synthetic and real data experiments in various works (Vovk and Petej, 2012; Nouretdinov et al., 2018; Johansson et al., 2019b;a; 2023; van der Laan and Alaa, 2024; Vovk et al., 2005; Lei et al., 2018; Romano et al., 2019; Boström and Johansson, 2020; Romano et al., 2020; Gibbs et al., 2023). In this section, we evaluate two novel instances of these methods: CP using Venn-Abers calibration with the quantile loss (Section 4.1) and Venn multicalibration for regression using the squared error loss.

## 5.1. Venn-Abers conformal quantile calibration

We evaluate conformal prediction intervals constructed using Venn-Abers quantile calibration on real datasets, including the Medical Expenditure Panel Survey (MEPS) dataset (Cohen et al., 2009; MEPS, 2021), as well as the *Concrete*, *Community*, *STAR*, *Bike*, and *Bio* datasets from Romano et al. (2019), which are available in the cqr package. Each dataset is split into a training set (50%), a calibration set (30%), and a test set (20%). We implement Venn-Abers quantile calibration (**VA**) using absolute residual error as the conformity score and train the $1 - \alpha$ quantile model $f(\cdot)$ of the conformity score using xgboost (Chen and Guestrin, 2016). The baselines include **uncalibrated** intervals derived from $f(\cdot)$, a symmetric variant of conformalized quantile regression (**CQR**) (Romano et al., 2019), **Marginal** conformal prediction (**CP**) (Vovk et al., 2005; Lei et al., 2018), and Mondrian conformal prediction (**VM**) (Romano et al., 2020), with categories based on bins of the estimated $1 - \alpha$ quantiles. **VM** corresponds to **V**enn calibration with **M**ondrian histogram binning. For direct comparability, all baselines are based on the absolute residual error score $|y - \mu(x)|$, where $\mu(x)$ is a xgboost predictor of the conditional median of $y$, and intervals are thus centered around $\mu(x)$.

Averaged over 100 random data splits, Table 1 summarizes Monte Carlo estimates of marginal coverage, and conditional $\ell^1$-calibration error (CCE), and average interval width. For $f_n^*$, the isotonic calibration of $f$, the CCE is defined as

$$E_P \left[ \max\{0, P(Y \notin \widehat{C}_n(X) \mid f_n^*(X), \mathcal{C}_n) - \alpha\} \mid \mathcal{C}_n \right].$$

*Table 1.* Metrics for each dataset: Marginal Coverage, Conditional Calibration Error (CCE), and Average Width. For CCE, smaller values are preferred, and the minimum value for each dataset is bolded. For coverage, values close to 90% are desired, and the average width is ideally minimized while retaining coverage.

| Method | Bike | Bio | Star | Meps | Conc | Com |
|---|---|---|---|---|---|---|
| **Marginal Coverage** | | | | | | |
| Uncalibrated | 0.81 | 0.85 | 0.80 | 0.86 | 0.71 | 0.74 |
| Venn-Abers | 0.90 | 0.90 | 0.90 | 0.90 | 0.90 | 0.90 |
| CQR | 0.90 | 0.90 | 0.90 | 0.90 | 0.90 | 0.90 |
| Marginal | 0.90 | 0.90 | 0.90 | 0.90 | 0.90 | 0.90 |
| VM (5 bin) | 0.90 | 0.90 | 0.89 | 0.90 | 0.89 | 0.90 |
| VM (10 bin) | 0.90 | 0.90 | 0.89 | 0.90 | 0.88 | 0.89 |
| **Conditional Calibration Error (CCE)** | | | | | | |
| Uncalibrated | 0.11 | 0.088 | 0.11 | 0.053 | 0.20 | 0.17 |
| Venn-Abers | **0.019** | **0.017** | 0.024 | **0.018** | **0.035** | **0.028** |
| CQR | 0.031 | 0.020 | 0.026 | 0.020 | 0.037 | 0.031 |
| Marginal | 0.10 | 0.053 | **0.020** | 0.052 | 0.057 | 0.058 |
| VM (5 bin) | 0.033 | 0.025 | 0.023 | 0.026 | 0.044 | 0.030 |
| VM (10 bin) | 0.022 | 0.020 | 0.028 | 0.022 | 0.049 | 0.030 |
| **Average Width** | | | | | | |
| Uncalibrated | 83 | 12 | 620 | 2.4 | 9.4 | 0.22 |
| Venn-Abers | 100 | 14 | 780 | 2.8 | 17 | 0.40 |
| CQR | 98 | 14 | 780 | 2.7 | 16 | 0.38 |
| Marginal | 140 | 15 | 780 | 2.9 | 18 | 0.46 |
| VM (5 bin) | 99 | 14 | 770 | 2.8 | 16 | 0.38 |
| VM (10 bin) | 100 | 14 | 770 | 2.8 | 16 | 0.38 |

All calibrated methods achieve adequate marginal coverage, as guaranteed by theory, while the uncalibrated intervals exhibit poor coverage. **VA** consistently achieves the lowest or comparable CCE across datasets, as expected from Theorems 3.4 and 4.1, outperforming or matching the state-of-the-art **CQR** in terms of coverage, CCE, and width. Although **VM** CP improves with more bins, its CCE remains higher than that of **VA**, highlighting the advantage of data-adaptive binning via isotonic regression.

## 5.2. Venn mean multicalibration

We evaluate Venn multicalibration for regression with squared error loss on the same datasets as in the previous experiment. To our knowledge, there is no prior work on set multicalibrators, and thus no existing comparators to Algorithm 3. Accordingly, the primary goal of this experiment is to assess the quality of the set-valued predictions in terms of (i) their size and (ii) the calibration error of the oracle multicalibrated prediction that is guaranteed to lie within the set.

Each dataset is split into a training set (40%), a calibration set (40%), and a test set (20%). We train the model $f$ using median regression with xgboost, such that the model is miscalibrated for the mean when the outcomes are skewed.

We apply Alg. 3 with $\mathcal{G}$ defined as the linear span of an additive spline basis of the features, aiming for multicalibration over additive functions. Specifically, for continuous features, we generate cubic splines with five knot points, and for categorical features, we apply one-hot encoding. As baselines, we consider the uncalibrated model and the point-calibrated model obtained by adjusting $f$ via offset linear regression on $\mathcal{C}_n$ based on $\mathcal{G}$. All outcomes are rescaled to lie in $[0, 1]$ for comparability across datasets.

Averaged averaged over 100 random data splits, Table 2 summarizes the sample size $(n)$, feature dimension $(p)$, conditional multicalibration errors for the uncalibrated, point calibrated, and oracle Venn-calibrated predictions, and the average Venn prediction set width. The oracle Venn-calibrated prediction is the marginally perfectly calibrated prediction $f_{n+1}^{(X_{n+1}, Y_{n+1})}$ necessarily contained in the prediction set. For a basis $\{b_j(\cdot)\}_{j=1}^m$ of $\mathcal{G}$, the multicalibration error of a model $\widehat{f}$ is defined as the $\ell^2$ norm of the test-set in-sample calibration errors:

$$\left\{ \frac{1}{n_{\text{test}}} \sum_{i=1}^{n_{\text{test}}} b_j(X_i)\{Y_i - \widehat{f}(X_i)\} : j \in [m] \right\}.$$

This error quantifies how well the model satisfies the multicalibration criterion across a rich collection of (potentially overlapping) subpopulations defined by the functions $b_j$, as defined in (5). In particular, it captures calibration error both for discrete subgroups (e.g., based on binary covariates) and for continuous covariates through smooth density ratio weights.

*Table 2.* Sample size and feature dimension $(n, p)$, calibration errors for the uncalibrated model, calibrated model, Venn-calibrated model, and mean prediction set width for each dataset.

|      | Dim. | \multicolumn{3}{c}{Calibration Error} | Width |
|------|------|-------|-------|-------|-------|
|      | $(n, p)$ | Uncal | Calibr | Venn | Venn |
| Bike | (4354, 18)  | 0.0019 | 0.0015 | **0.0015** | 0.0086 |
| Bio  | (18292, 9)  | **0.0073** | 0.0100 | 0.0094 | 0.0100 |
| Star | (864, 39)   | 0.0098 | 0.0113 | **0.0078** | 0.3260 |
| Meps | (6262, 139) | 0.0032 | 0.0017 | **0.0016** | 0.0088 |
| Conc | (412, 8)    | 0.0077 | 0.0081 | **0.0064** | 0.1430 |
| Comm | (797, 101)  | 0.0099 | 0.0209 | **0.0055** | 0.6650 |

The oracle Venn-calibrated model consistently achieves smaller calibration errors than the point-calibrated model across all datasets and outperforms the uncalibrated model in all but one dataset. Its improvement is more pronounced in settings with wider Venn prediction sets, which correspond to smaller effective sample sizes $\frac{n}{p}$. In these cases, naive multicalibration is more variable and prone to overfitting. This aligns with expectations, as wider prediction sets reflect greater uncertainty in the finite-sample calibration of point-calibrated predictions.

## Impact Statement

This paper presents work whose goal is to advance the field of Machine Learning. There are many potential societal consequences of our work, none which we feel must be specifically highlighted here.

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

## A. Code Availability

Python code implementing *Venn-Abers* and *Venn multicalibration* methods for both squared error and quantile losses is available in the `VennCalibration` package at the following GitHub repository:



https://github.com/Larsvanderlaan/VennCalibration



The repository includes scripts and documentation for reproducing all experiments in this paper.

## B. Background on isotonic calibration

Isotonic calibration (Zadrozny and Elkan, 2002; Niculescu-Mizil and Caruana, 2005) is a data-adaptive histogram binning method that learns the bins using isotonic regression, a nonparametric method traditionally used for estimating monotone functions (Barlow and Brunk, 1972; Groeneboom and Lopuhaa, 1993). Specifically, the bins are selected by minimizing an empirical MSE criterion under the constraint that the calibrated predictor is a non-decreasing monotone transformation of the original predictor. Isotonic calibration is motivated by the heuristic that, for a good predictor $f$, the calibration function $\theta_{P,f}$ should be approximately monotone as a function of $f$. For instance, when $f(\cdot) = E_P[Y \mid X = \cdot]$, the mapping $f \mapsto \theta_{P,f} = f$ is the identity function. Isotonic calibration is distribution-free — it does not rely on monotonicity assumptions — and, in contrast with histogram binning, it is tuning parameter-free and naturally preserves the mean-square error of the original predictor (as the identity transform is monotonic) (van der Laan et al., 2023).

For clarity, we focus on the regression case where $\ell$ denotes the squared error loss. Formally, isotonic calibration takes a predictor $f$ and a calibration dataset $\mathcal{C}_n$ and produces the calibrated model $f_n^* := \theta_n \circ f$, where $\theta_n : \mathbb{R} \to \mathbb{R}$ is an isotonic step function obtained by solving the optimization problem:

$$\theta_n \in \operatorname*{argmin}_{\theta \in \Theta_{\text{iso}}} \sum_{i=1}^n \left\{Y_i - \theta(f(X_i))\right\}^2, \tag{6}$$

where $\Theta_{\text{iso}}$ denotes the set of all univariate, piecewise constant functions that are monotonically nondecreasing. Following Groeneboom and Lopuhaa (1993), we consider the unique càdlàg piecewise constant solution to the isotonic regression problem, which has jumps only at observed values in $\{f(X_i) : i \in [n]\}$. The first-order optimality conditions of the convex optimization problem imply that the isotonic solution $\theta_n$ acts as a binning calibrator with respect to a data-adaptive set of bins determined by the jump points of the step function $\theta_n$. Thus, isotonic calibration provides perfect in-sample calibration. Specifically, for any transformation $g : \mathbb{R} \to \mathbb{R}$, the perturbed step function $\varepsilon \mapsto \theta_n + \varepsilon(g \circ \theta_n)$ remains isotonic for all sufficiently small $\varepsilon$ such that $|\varepsilon| \sup_{t \in f(\mathcal{X})} |(g \circ \theta_n)(t)|$ is less than the maximum jump size of $\theta_n$, given by $\sup_{t \in f(\mathcal{X})} |\theta_n(t) - \theta_n(t-)|$. Since $\theta_n$ minimizes the empirical mean square error criterion over all isotonic functions, it follows that, for each function $g : \mathbb{R} \to \mathbb{R}$, the following condition holds:

$$\frac{d}{d\varepsilon} \frac{1}{2} \sum_{i=1}^n \left\{Y_i - \theta_n(f(X_i)) - \varepsilon g(\theta_n(f(X_i)))\right\}^2 \bigg|_{\varepsilon=0} = \sum_{i=1}^n g(f_n^*(X_i))\{Y_i - f_n^*(X_i)\} = 0.$$

These orthogonality conditions are equivalent to perfect in-sample calibration. In particular, by taking $g$ as the level set indicator $t \mapsto \mathbb{I}(t = \theta_n(f(x)))$, we conclude that the isotonic calibrated predictor $f_n^*$ is in-sample calibrated.

## C. Proofs

### C.1. Proofs for Venn calibration

*Proof of Theorem 3.1.* From C3, we know that

$$\sum_{i=1}^n \mathbb{1}\{f_{n+1}^*(X_i) = f_{n+1}^*(x)\}\partial\ell(f_{n+1}^*(X_i), Z_i) = 0.$$

This condition implies, for every transformation $g : \mathbb{R} \to \mathbb{R}$, that

$$\sum_{i=1}^n g(f_{n+1}^*(X_i))\partial\ell(f_{n+1}^*(X_i), Z_i) = 0.$$

Taking the expectation of both sides, we find that

$$0 = \mathbb{E}\left[\sum_{i=1}^{n} g(f_{n+1}^*(X_i))\partial\ell(f_{n+1}^*(X_i), Z_i)\right]$$
$$= \sum_{i=1}^{n} \mathbb{E}\left[g(f_{n+1}^*(X_i))\partial\ell(f_{n+1}^*(X_i), Z_i)\right].$$

Note that $f_{n+1}^*$ is trained on all of $\mathcal{C}_{n+1}^*$ and is thus invariant to permutations of $\{(X_i, Y_i)\}_{i=1}^{n+1}$. Since $\{(X_i, Y_i)\}_{i=1}^{n+1}$ are exchangeable by C1, it follows that $g(f_{n+1}^*(X_i))\partial\ell(f_{n+1}^*(X_i), Z_i)$ is exchangeable over $i \in [n+1]$. Thus, the previous display implies, for every transformation $g : \mathbb{R} \to \mathbb{R}$, that

$$0 = \sum_{i=1}^{n} \mathbb{E}\left[g(f_{n+1}^*(X_{n+1}))\partial\ell(f_{n+1}^*(X_{n+1}), Z_{n+1})\right]$$
$$0 = \mathbb{E}\left[g(f_{n+1}^*(X_{n+1}))\partial\ell(f_{n+1}^*(X_{n+1}), Z_{n+1})\right]$$
$$0 = \mathbb{E}\left[g(f_{n+1}^*(X_{n+1}))\mathbb{E}[\partial\ell(f_{n+1}^*(X_{n+1}), Z_{n+1}) \mid f_{n+1}^*(X_{n+1})]\right],$$

where the final equality follows from the law of total expectation. Taking $g$ such that $g(f_{n+1}^*(x)) = \mathbb{E}[\partial\ell(f_{n+1}^*(X_{n+1}), Z_{n+1}) \mid f_{n+1}^*(X_{n+1}) = f_{n+1}^*(x)]$, which exists by C2, we conclude that

$$\mathbb{E}\left[\left\{\mathbb{E}[\partial\ell(f_{n+1}^*(X_{n+1}), Z_{n+1}) \mid f_{n+1}^*(X_{n+1})]\right\}^2\right] = 0.$$

$\square$

*Proof of Theorem 3.2*. For a uniformly bounded function class $\mathcal{F}$, let $N(\epsilon, \mathcal{F}, L_2(P))$ denote the $\epsilon-$covering number (**?**) of $\mathcal{F}$ with respect to $L_2(P)$ and define the uniform entropy integral of $\mathcal{F}$ by

$$\mathcal{J}(\delta, \mathcal{F}) := \int_0^\delta \sup_Q \sqrt{\log N(\epsilon, \mathcal{F}, L_2(Q))}\, d\epsilon,$$

where the supremum is taken over all discrete probability distributions $Q$. For two quantities $x$ and $y$, we use the expression $x \lesssim y$ to mean that $x$ is upper bounded by $y$ times a universal constant that may only depend on global constants that appear in our conditions.

We know that $\theta_n^{(x,y)}$ almost surely belongs to a uniformly bounded function class $\mathcal{F}_n$ consisting of 1D functions with at most $k(n)$ constant segments. Then, $\mathcal{F}_n$ has finite uniform entropy integral with $\mathcal{J}(\delta, \mathcal{F}_n) \lesssim \delta\sqrt{k(n)\log(1/\delta)}$. Define $\mathcal{F}_{f,n} := \{\theta \circ f : \theta \in \mathcal{F}_n\}$. We claim that $\mathcal{J}(\delta, \mathcal{F}_{f,n}) \lesssim \delta\sqrt{k(n)\log(1/\delta)}$. This follows since, by the change-of-variables formula,

$$\mathcal{J}(\delta, \mathcal{F}_{f,n}) = \int_0^\delta \sup_Q \sqrt{N(\varepsilon, \mathcal{F}_{f,n}, \|\cdot\|_Q)}\, d\varepsilon$$
$$= \int_0^\delta \sup_Q \sqrt{N(\varepsilon, \mathcal{F}_n, \|\cdot\|_{Q\circ f})}\, d\varepsilon$$
$$= \mathcal{J}(\delta, \mathcal{F}_n).$$

where, with a slight abuse of notation, $Q \circ f$ is the push-forward probability measure for the random variable $f(X)$.

By assumption, we have perfect in-sample calibration: for all $g : \mathbb{R} \to \mathbb{R}$,

$$\sum_{i=1}^{n} g(\theta_n^{(x,y)}(f(X_i)))\partial\ell(\theta_n^{(x,y)}(f(X_i), Y_i)) + g(\theta_n^{(x,y)}(f(x)))\partial\ell(\theta_n^{(x,y)}(f(x)), y) = 0.$$

Take $g$ such that $g \circ \theta_n^{(x,y)}$ equals $t \mapsto E_P[\partial \ell(\theta_n^{(x,y)}(f(x)), y) \mid \theta_n^{(x,y)}(f(X)) = t, \mathcal{C}_n]$. Then, denoting $\gamma_f(\theta_n^{(x,y)}, \cdot) : x' \mapsto E_P[\partial \ell(\theta_n^{(x,y)}(f(x')), z') \mid \theta_n^{(x,y)}(f(X)) = \theta_n^{(x,y)}(f(x')), \mathcal{C}_n]$, we find that

$$\sum_{i=1}^n \gamma_f(\theta_n^{(x,y)}, X_i) \partial \ell(\theta_n^{(x,y)}(f(X_i), Y_i)) + \gamma_f(\theta_n^{(x,y)}, x) \partial \ell(\theta_n^{(x,y)}(f(x)), y) = 0.$$

By assumption, $\gamma_f(\theta_n^{(x,y)}, X) \partial \ell(\theta_n^{(x,y)}(f(x)), y)$ is uniformly bounded, such that

$$\frac{1}{n} \sum_{i=1}^n \gamma_f(\theta_n^{(x,y)}, X_i) \partial \ell(\theta_n^{(x,y)}(f(X_i), Y_i)) = O(n^{-1}).$$

Adding and subtracting, we have that

$$P_n \gamma_f(\theta_n^{(x,y)}, \cdot) \partial \ell(\theta_n^{(x,y)}(f(\cdot), \cdot)) = O(n^{-1})$$
$$P \gamma_f(\theta_n^{(x,y)}, \cdot) \partial \ell(\theta_n^{(x,y)}(f(\cdot), \cdot)) + (P_n - P) \gamma_f(\theta_n^{(x,y)}, \cdot) \partial \ell(\theta_n^{(x,y)}(f(\cdot), \cdot)) = O(n^{-1})$$
$$P \{\gamma_f(\theta_n^{(x,y)}, \cdot)\}^2 + (P_n - P) \gamma_f(\theta_n^{(x,y)}, \cdot) \partial \ell(\theta_n^{(x,y)}(f(\cdot), \cdot)) = O(n^{-1}),$$

where, in the final equality, we used that $P \gamma_f(\theta_n^{(x,y)}, \cdot) \partial \ell(\theta_n^{(x,y)}(f(\cdot), \cdot)) = P\{\gamma_f(\theta_n^{(x,y)}, \cdot)\}^2$ by the law of total expectation.

The random quantity we wish to bound by $\widehat{\delta}_n^2 P\{\gamma_f(\theta_n^{(x,y)}, \cdot)\}^2$. Then, the previous display implies

$$\widehat{\delta}_n^2 \leq \sup_{\theta \in \mathcal{F}_n : \|\gamma_f(\theta, \cdot)\|_P \leq \widehat{\delta}_n} |(P_n - P) \gamma_f(\theta, \cdot) \partial \ell(\theta(f(\cdot), \cdot))| + O(n^{-1}).$$

By boundedness of $\partial \ell(\theta(f(\cdot), \cdot))$, $\|\gamma_f(\theta, \cdot) \partial \ell(\theta(f(\cdot), \cdot))\|_P \leq K \|\gamma_f(\theta, \cdot)\|_P$ for some $K < \infty$. Thus,

$$\widehat{\delta}_n^2 \leq \sup_{g \in \mathcal{G}_f : \|g\|_P \leq K\widehat{\delta}_n} |(P_n - P) g| + O(n^{-1}),$$

where $\mathcal{G}_f := \{g_1 g_2 : g_1 \in \mathcal{G}_{1,f}, g_2 \in \mathcal{G}_{2,f}\}$ with $\mathcal{G}_{1,f} := \{\partial \ell(\theta(f(\cdot), \cdot)) : \theta \in \mathcal{F}_n\}$ and $\mathcal{G}_{2,f} := \{\gamma_f(\theta, \cdot) : \theta \in \mathcal{F}_n\}$.

We claim that $\mathcal{J}(\delta, \mathcal{G}_f) \lesssim \mathcal{J}(\delta, \mathcal{F}_n) \lesssim \delta \sqrt{k(n) \log(1/\delta)}$ By assumption, the following Lipschitz condition holds almost surely: $|\partial \ell(\theta_1(f(X)), Y) - \partial \ell(\theta_2(f(X)), Y)| \lesssim |\theta_1(f(X)) - \theta_2(f(X))|$. It follows that $\mathcal{J}(\delta, \mathcal{G}_{1,f}) \lesssim \mathcal{J}(\delta, \mathcal{F}_{f,n})$. Moreover, $\mathcal{J}(\delta, \mathcal{G}_{2,f}) \lesssim \mathcal{J}(\delta, \mathcal{F}_{f,n})$, since $\gamma_f(\theta, \cdot) \in \mathcal{F}_{f,n}$ is a piecewise constant function with at most $k(n)$ constant segments for each $\theta \in \mathcal{F}_n$. Therefore, $\mathcal{J}(\delta, \mathcal{G}_f) \lesssim \mathcal{J}(\delta, \mathcal{F}_{f,n}) \lesssim \mathcal{J}(\delta, \mathcal{F}_n)$ and the claim follows.

Define $\phi_n(\delta) := \sup_{g \in \mathcal{G}_f : \|g\|_P \leq K\delta} |(P_n - P) g|$. Then, we can write

$$\widehat{\delta}_n^2 \leq \phi_n(\widehat{\delta}_n) + O(n^{-1}).$$

Applying Theorem 2.1 in Van Der Vaart and Wellner (2011), we have that for any $\delta$ satisfying $\sqrt{n} \delta^2 \gtrsim \mathcal{J}(\delta, \mathcal{G}_f)$,

$$\mathbb{E}[\phi_n(\delta)] \lesssim n^{-\frac{1}{2}} \mathcal{J}(\delta, \mathcal{G}_f).$$

Consequently, since $\mathcal{J}(\delta, \mathcal{G}_f) \lesssim \mathcal{J}(\delta, \mathcal{F}_n) \lesssim \delta \sqrt{k(n) \log(1/\delta)}$, it follows that for any $\delta \geq \sqrt{\frac{k(n) \log(1/\delta)}{n}}$,

$$\mathbb{E}[\phi_n(\delta)] \lesssim \delta \sqrt{k(n) \log(1/\delta)/n}.$$

It can be shown that $\delta_n^2 := k(n) \log(n/k(n))$ satisfies the critical inequality $\delta_n \geq \sqrt{\frac{k(n) \log(1/\delta_n)}{n}}$, such that the previous identifies can be applied with $\delta := \delta_n$. Showing the asserted stochastic order, $\widehat{\delta}_n^2 = O_p(\delta_n^2)$ with $\delta_n^2 := k(n) \log(n/k(n))$, is equivalent to demonstrating that for all $\epsilon > 0$, there exists a sufficiently large $2^S$ such that

$$\limsup_{n \to \infty} \mathbb{P}(\delta_n^{-2} \widehat{\delta}_n^2 > 2^S) < \epsilon.$$

To this end, we need to show $\lim_{n\to\infty} \mathbb{P}(\delta_n^{-2}\widehat{\delta}_n^2 > 2^S) \to 0$ as $S \to \infty$. Define the event $A_s := \left\{\delta_n^{-2}\widehat{\delta}_n^2 \in (2^s, 2^{s+1}]\right\}$ for each $s$. Using a peeling argument and Markov's inequality, we obtain

$$\mathbb{P}\big(\delta_n^{-2}\widehat{\delta}_n^2 > 2^S\big) \leq \sum_{s=S}^{\infty} \mathbb{P}\big(2^{s+1} \geq \delta_n^{-2}\widehat{\delta}_n^2 > 2^s\big)$$

$$\leq \sum_{s=S}^{\infty} \mathbb{P}\big(A_s, \widehat{\delta}_n^2 \leq \phi_n(\widehat{\delta}_n) + O(n^{-1})\big)$$

$$\leq \sum_{s=S}^{\infty} \mathbb{P}\big(A_s, \widehat{\delta}_n^2 \leq \phi_n(\widehat{\delta}_n) + O(n^{-1})\big)$$

$$\leq \sum_{s=S}^{\infty} \mathbb{P}\big(\delta_n^2 2^s < \widehat{\delta}_n^2 \leq \phi_n\big(\delta_n 2^{\frac{s+1}{2}}\big) + O(n^{-1})\big)$$

$$\leq \sum_{s=S}^{\infty} \mathbb{P}\big(\delta_n^2 2^s < \phi_n\big(\delta_n 2^{\frac{s+1}{2}}\big) + O(n^{-1})\big)$$

$$\leq \sum_{s=S}^{\infty} \frac{\mathbb{E}\big[\phi_n\big(\delta_n 2^{\frac{s+1}{2}}\big)\big] + O(n^{-1})}{\delta_n^2 2^s} \leq \sum_{s=S}^{\infty} \frac{n^{-1/2}\delta_n 2^{\frac{s+1}{2}} \sqrt{k(n)\log(n/k(n))} + O(n^{-1})}{\delta_n^2 2^s}$$

$$\leq \sum_{s=S}^{\infty} \frac{2^{\frac{s+1}{2}} + O(1)}{2^s} \to_{S\to\infty} 0.$$

Thus, $\widehat{\delta}_n^2 = O_p(\delta_n^2)$ and the result follows.

$\square$

*Proof of Theorem 3.4 .* This proof follows from a generalization of the proofs of Theorem 1 for treatment effect calibration and propensity score calibration in van der Laan et al. (2023) and van der Laan et al. (2024a).

Recall that $f_n^{(x,y)} = \theta_n^{(x,y)} \circ f$. Under C8, up to a change of notation, the proof of Lemma 3 establishes that the map $t \mapsto E_P[\partial\ell(f_n^{(x,y)}(X), Z) \mid f_n^{(x,y)}(X) = \theta_n^{(x,y)}(t), \mathcal{C}_n]$ has a total variation norm almost surely bounded by $3B$. Consequently, the function $\gamma_f(\theta_n^{(x,y)}, \cdot) : x \mapsto E_P[\partial\ell(f_n^{(x,y)}(X), Z) \mid f_n^{(x,y)}(X) = f_n^{(x,y)}(x), \mathcal{C}_n]$ is a transformation of $f$ with a total variation norm almost surely bounded by $3B$.

Let $\mathcal{F}_{TV}$ denote the space of 1D functions with total variation norm bounded by $3B$. Let $\mathcal{F}_{iso}$ denote the space of isotonic functions that are uniformly bounded, such that the isotonic regression solution $\theta_n^{(x,y)}$ belongs to this set. Note that $\mathcal{J}(\delta, \mathcal{F}_{TV}) \lesssim \sqrt{\delta}$ and $\mathcal{J}(\delta, \mathcal{F}_{iso}) \lesssim \sqrt{\delta}$.

Proceeding exactly as in the proof of Theorem 3.2, we can show that the quantity we wish to bound, $\widehat{\delta}_n^2 := P\{\gamma_f(\theta_n^{(x,y)}, \cdot)\}^2$, satisfies

$$\widehat{\delta}_n^2 \leq \sup_{\theta \in \mathcal{F}_n : \|\gamma_f(\theta, \cdot)\|_P \leq \widehat{\delta}_n} |(P_n - P)\gamma_f(\theta, \cdot)\partial\ell(\theta(f(\cdot), \cdot))| + O(n^{-1}).$$

By the boundedness of $\partial\ell(\theta(f(\cdot), \cdot))$, we have $\|\gamma_f(\theta, \cdot)\partial\ell(\theta(f(\cdot), \cdot))\|_P \leq K\|\gamma_f(\theta, \cdot)\|_P$ for some $K < \infty$. Thus,

$$\widehat{\delta}_n^2 \leq \sup_{g \in \mathcal{G}_f : \|g\|_P \leq K\widehat{\delta}_n} |(P_n - P)g| + O(n^{-1}),$$

where $\mathcal{G}_f := \{g_1 g_2 : g_1 \in \mathcal{G}_{1,f}, g_2 \in \mathcal{G}_{2,f}\}$ with $\mathcal{G}_{1,f} := \{\partial\ell(\theta(f(\cdot), \cdot)) : \theta \in \mathcal{F}_{TV}\}$ and $\mathcal{G}_{2,f} := \{\gamma_f(\theta, \cdot) : \theta \in \mathcal{F}_{iso}\}$. An argument similar to the proof of Theorem 3.2 shows that $\mathcal{J}(\delta, \mathcal{G}_f) \lesssim \mathcal{J}(\delta, \mathcal{F}_{TV}) + \mathcal{J}(\delta, \mathcal{F}_{iso}) \lesssim \sqrt{\delta}$.

The result now follows by applying an argument identical to the proof of Theorem 3.2, where we set $\delta_n^2 := n^{-2/3}$ and use $\mathcal{J}(\delta, \mathcal{G}_f) \lesssim \sqrt{\delta}$.

$\square$

## C.2. Proofs for Venn multicalibration

*Proof of Theorem 3.5.* By C9, we have almost surely for each $g \in \mathcal{G}$ that

$$\sum_{i=1}^{n+1} \frac{\partial}{\partial t} \ell((f_{n+1}^* + tg)(X_i), Z_i)\Big|_{t=0} = 0.$$

Taking the expectations of both sides above and leveraging C1, we have

$$\mathbb{E}\left[\sum_{i=1}^{n+1} \frac{\partial}{\partial t} \ell((f_{n+1}^* + tg)(X_i), Z_i)\Big|_{t=0}\right] = 0,$$

$$\sum_{i=1}^{n+1} \mathbb{E}\left[\frac{\partial}{\partial t} \ell((f_{n+1}^* + tg)(X_i), Z_i)\Big|_{t=0}\right] = 0,$$

$$\sum_{i=1}^{n+1} \mathbb{E}\left[\frac{\partial}{\partial t} \ell((f_{n+1}^* + tg)(X_{n+1}), Z_{n+1})\Big|_{t=0}\right] = 0,$$

$$\mathbb{E}\left[\frac{\partial}{\partial t} \ell((f_{n+1}^* + tg)(X_{n+1}), Z_{n+1})\Big|_{t=0}\right] = 0,$$

as desired. $\qquad\square$

*Proof of Corollary 3.6.* This result is a direct consequence of Theorem 3.5, but we provide an independent proof for clarity and completeness.

Define $f_{n+1}^* := f_n^{(X_{n+1}, Y_{n+1})}$, and note that $f_{n+1}^*(X_{n+1})$ is an element of $f_{n,X_{n+1}}(X_{n+1})$ by construction. The first-order optimality conditions of the empirical risk minimizer $g_n^{(X_{n+1}, Y_{n+1})}$ imply that, for each $g \in \mathcal{G}$,

$$\frac{1}{n+1} \sum_{i=1}^{n+1} g(X_i) \left\{ Y_i - f_{n+1}^*(X_i) \right\} = 0.$$

Taking the expectation of both sides, which we can do by C2, and leveraging C1, we find that

$$0 = \frac{1}{n+1} \sum_{i=1}^{n+1} \mathbb{E}\left[g(X_i) \left\{ Y_i - f_{n+1}^*(X_i) \right\}\right]$$

$$= \frac{1}{n+1} \sum_{i=1}^{n+1} \mathbb{E}\left[g(X_{n+1}) \left\{ Y_{n+1} - f_{n+1}^*(X_{n+1}) \right\}\right]$$

$$= \mathbb{E}\left[g(X_{n+1}) \left\{ Y_{n+1} - f_{n+1}^*(X_{n+1}) \right\}\right].$$

$\qquad\square$

## C.3. Proofs for conformal prediction

*Proof.* Proof of Theorem 4.1 Under the assumption that $S_i \neq f_{n+1}^*(X_i)$ almost surely for all $i \in [n+1]$, it is shown in Section 2.2 of Gibbs et al. (2023) that the quantile loss is differentiable almost surely. Moreover, its derivative is given by

$$\partial \ell_\alpha(f(x), S(z)) = (1 - \alpha) - \mathbb{1}\{S(x) \geq f(x)\}.$$

The result now follows by application of Theorem 3.1. $\qquad\square$

*Proof.* Proof of Theorem 4.2 Under the assumption that $S_i \neq f_{n+1}^*(X_i)$ almost surely for all $i \in [n+1]$, it is shown in Section 2.2 of Gibbs et al. (2023) that the quantile loss is differentiable almost surely. Moreover, its derivative is given by

$$\frac{d}{d\varepsilon} \ell_\alpha((f + g\varepsilon)(x), S(z))\big|_{\varepsilon=0} = g(x) \left[(1 - \alpha) - \mathbb{1}\{S(x) \geq f(x)\}\right].$$

The result now follows by application of Theorem 3.5. $\qquad\square$

