# OpenReview forum: "Generalized Venn and Venn-Abers Calibration with Applications in Conformal Prediction"
_ICML.cc/2025/Conference — ICML 2025 poster_

### Official Review · Reviewer_jn2h · 2025-03-11

**Overall Recommendation:** 3

**Summary:**

This paper proposes  Venn and Venn-Abers Calibration defined as connected to the loss function. First, the isotonic regression and quantile losses are examined for the marginal calibration. Furthermore, conditional calibration is discussed. There are a series of theoretical results for the proposed algorithms. Some experiments show the relative superiority of the proposed algorithm.

## update after rebuttal

**Claims And Evidence:**

Calibration is achieved by the post-processing type learning combined with augmentation. The theries seems sound and empirical results looks better.

**Essential References Not Discussed:**

None

**Experimental Designs Or Analyses:**

The experiments are limited, in my opinion, since some cases in theorems are not addressed thoroughly. For example, using histogram binning is not examined.  The details of datasets, such as the sizes, are not provided.

**Methods And Evaluation Criteria:**

The proposed algorithm looks sound, and the metrics are based on the proposed metrics.
If the authors consider other metrics used in the conventional metrics, such as the ECE (Guo et al. 2017), the performance can be compared to the conventional ones.

**Other Comments Or Suggestions:**

None

**Other Strengths And Weaknesses:**

Positive: The theoretical validations can be strong points, and the definition of calibration is general and has potential.

negative: The applications and comparisons with other definitions are limited.

**Questions For Authors:**

Q1: Let me know the merits or advantages of using augmented data in the proposed alg.
Q2: What are the representative losses for the proposed algorithm? Can you provide the ten of loss functions?

**Relation To Broader Scientific Literature:**

This study is closely related to the reliability or trust AI.

**Theoretical Claims:**

It is nice to present the series of theorems to validate the proposed algorithms.

---

> ### Author Rebuttal · Authors · 2025-04-01
>
> Q1: By augmenting the dataset with all possible imputed outcomes for the test point whose outcome we wish to predict we are able to convert a point prediction into set prediction, where the width of this set captures epistemic uncertainty in the calibration process. This augmentation approach is the workhorse of the finite-sample distribution-free calibration guarantees of the original Venn-Abers calibration and conformal prediction procedures of Vovk and similarly is a crucial component of our algorithms.
>
> Q2: The proposed algorithm applies to any smooth, bounded loss function, including standard choices such as squared error loss and quantile loss. It also accommodates more specialized losses, such as those used in missing data settings, as well as losses tailored to estimating conditional functionals like the conditional average treatment effect or the conditional relative risk, as discussed in several of our cited works. We will include additional examples of representative loss functions in the revised manuscript to clarify the range of settings where our method applies.

---

> > ### Comment · Reviewer_jn2h · 2025-04-09
> >
> > Thanks for your reply. Many points are resolved, and I'll keep my score.

---

### Official Review · Reviewer_dpJ8 · 2025-03-14

**Overall Recommendation:** 3

**Summary:**

This paper introduces a unified framework for Venn and Venn-Abers calibration, generalizing Venn calibration to hold with respect to arbitrary given loss functions. Unlike point calibrators (e.g., histogram binning, isotonic regression), which map predictions to a single calibrated value, Venn calibration constructs prediction sets that are guaranteed to contain at least one perfectly calibrated prediction in finite samples. The authors show how to set up a Venn calibrator for any loss to produce prediction sets with marginal and conditional calibration guarantees (where here the word "conditional" refers to conditioning on the calibration set). The marginal guarantees hold in finite samples, while the conditional guarantees hold in the large sample regime. In particular, using isotonic regression as the point calibrator underlying Venn calibration, the Venn-Abers predictor for general losses is obtained. The authors also propose Venn multicalibration, ensuring calibration across a given family subpopulations.

For quantile regression (where the loss is chosen to be the pinball loss), they show that Venn calibration aligns with conformal prediction, achieving quantile-conditional coverage, and that multicalibrated conformal prediction is a special case of Venn multicalibration --- thus encompassing some existing methods. In addition, several experiments are provided --- for regular multicalibration (i.e., for Brier score/L_2 loss) and for conformal multicalibration (i.e., for the pinball loss) that verify that these methods are efficiently implementable (at least for finite label spaces) and that their theoretical guarantees lead to reasonable calibration rates in practice.

#######
Update after rebuttal:

I have read the authors' response. It sufficiently addresses my questions, and I therefore keep my original score. I have also read the discussion with the other reviewers, and I appreciate the further future updates proposed by the authors, in particular the addition/refinement of experiments in the empirical part of the paper.

**Claims And Evidence:**

Yes, both theoretical and empirical claims are clearly supported in the paper.

**Essential References Not Discussed:**

N/A

**Experimental Designs Or Analyses:**

Yes, the experimental designs are simple and sound.

**Methods And Evaluation Criteria:**

The datasets make sense, having been derived e.g. from prior research on conformal prediction; the evaluation criteria based on calibration are also correct.

**Other Comments Or Suggestions:**

Several points could use, I think, some further elaboration.

First, could you comment on the Sherman-Morrison approach in the context of the implementation of the multicalibration algorithm?
Second, in a similar vein, when the case of infinite Y is mentioned in several instances, it is suggested that looking at the extreme values of y will help --- elaborating on this statement somewhat more formally would increase clarity.
Third, to provide context for the reader as to the extension of calibration to general losses, it may be helpful to elaborate on this in the introduction by placing this in the context of calibration for general elicitable properties (of which means and quantiles are examples). E.g. as observed in the cited paper Statistical Scope of Multicalibration, this relationship is tight (and they give point multicalibrators for all well-behaved losses in parallel to how Venn calibration is generalized here).

**Other Strengths And Weaknesses:**

As mentioned before, the strength of the paper is that it usefully generalizes previous fundamental set-based algorithms for calibration. While the formulation of these extensions to arbitrary loss functions is natural and appears straightforward, but the proofs for some of the properties of the generalized algorithms derived in this paper are not straightforward and require some care. E.g., conditions on underlying point calibrators for Venn calibration must be established, and verified for isotonic regression to give rise to Venn-Abers; and the conditional calibration Theorem 3.2 is nontrivial.

The experiments are relatively rudimentary, including few or outdated baselines (such as conformalized quantile regression on the conformal prediction end or one uncalibrated and one group calibrated predictor on the mean calibration end), which is the main weakness; however, viewing this paper as mostly theoretical, I don't consider this to be a big flaw.

**Questions For Authors:**

Other than the above mentioned concern that the empirics are not very extensive, I don't have major complaints. A substantively strengthened empirical section (with more baselines, more nonconformity scores, a clear discussion of the differences of the implementation details to existing multicalibration methods, and some further substantiation of the practical feasibility of these methods for infinite Y) would further boost my evaluation of this paper, but that is not strictly required.

**Relation To Broader Scientific Literature:**

This work provides a clear generalization of preceding work on calibration and multicalibration, extending so-called Venn and Venn-Abers predictors, which originally worked for the calibrated mean setting (that corresponds to L2 loss) to arbitrary well-behaved losses. Each loss induces its own notion of calibration, and in particular for L2 and pinball loss, the Venn and Venn-Abers procedures (and their respective contextual strengthenings) give rise to mean and conformal calibration and multicalibration algorithms similar (but sometimes different) than the ones which have been obtained in prior work.

**Theoretical Claims:**

I read all of the proofs in the appendix moderately carefully, and believe them to be generally correct (up to a few technical details that weren't exhaustively checked). (The one instance of a "typo" is where the change-of-variables formula is omitted in line 690 in the Supplementary material.)

---

> ### Author Rebuttal · Authors · 2025-04-01
>
> Thank you for the helpful comments.
>
> - Thank you for these thoughtful suggestions. We will add clarification on how the Sherman–Morrison approach can be used to efficiently implement the algorithm. We appreciate the request for more detail on the case of infinite $\mathcal{Y}$, and will revise the relevant sections to more formally explain how analyzing extreme values of $y$ can suffice to determine the range of the Venn prediction set. We also agree that it would be helpful to situate our generalization of calibration within the broader context of elicitable properties. In the revised introduction, we will add background on this connection and clarify how calibration for general losses relates to elicitable properties.
>
> - We will add further details on the computational and implementation aspects of our approach, including how to efficiently handle infinite $\mathcal{Y}$ and how discretizing the model output can be used to approximate Venn–Abers calibration effectively.

---

> > ### Comment · Reviewer_dpJ8 · 2025-04-09
> >
> > Thank you for your reply! The proposed updates sound good, and I'll keep my score.

---

### Official Review · Reviewer_BPmy · 2025-03-14

**Overall Recommendation:** 4

**Summary:**

The authors propose a unified framework for Venn and Venn-Abers calibration that leverages binning calibrators to construct prediction sets that contain at least one marginally perfect calibrated prediction. Furthermore, they propose a novel method for Venn calibration technique across subpopulations. Their method outperforms the baselines in all but one case.

## Update after rebuttal
We have read the authors' responses, and they have addressed our concerns and promised to update the manuscript accordingly in the revised version. We lean toward acceptance and have updated our score to Accept.

**Claims And Evidence:**

(1)- The authors propose Venn multicalibration, ensuring finite-sample calibration across subpopulations. While they show how based on some preconditions (exchangeability and finite variance) and applying a particular algorithm, the multicalibration can be achieved for Venn calibrators. Neverthelesss, while the authors provide some empirical evidence on this aspect of their research, we consider the experiments should modified to strengthen their claims. In particular, multicalibration requires a calibrator to be calibrated across subpopulations, but the authors fail to identify subpopulations in each of the proposed datasets and to provide insights regarding the quality of calibration for each of those subpopulations.

**Essential References Not Discussed:**

We consider the following works to be relevant to the manuscript: (a) Gohar, U., & Cheng, L. (2023). A survey on intersectional fairness in machine learning: Notions, mitigation, and challenges. arXiv preprint arXiv:2305.06969.; (b) Silva Filho, T., Song, H., Perello-Nieto, M., Santos-Rodriguez, R., Kull, M., & Flach, P. (2023). Classifier calibration: a survey on how to assess and improve predicted class probabilities. Machine Learning, 112(9), 3211-3260.; and (c) Toccaceli, P. (2021). Conformal and Venn Predictors for large, imbalanced and sparse chemoinformatics data (Doctoral dissertation, Royal Holloway, University of London).

**Experimental Designs Or Analyses:**

(1)- While multicalibration requires a calibrator to be calibrated across subpopulations, the authors fail to identify subpopulations in each of the proposed datasets. We consider this to be a critical issue to provide insights regarding the quality of calibration for each of those subpopulations.

**Methods And Evaluation Criteria:**

(1)- While the authors consider several datasets that have been used in reputed works related to conformal predictions, we would appreciate it if they could ground their choice from the subpopulations perspective. The fact that the datasets present subpopulations is key to their experiments for multicalibration and datasets failing to satisfy that characteristic should be excluded from the multicalibration experiments.

(2)- We would appreciate it if the authors would justify their choice regarding baseline methods, to understand why these are relevant to this particular experiments. In particular, we miss comparisons against multicalibration techniques. While the authors reference some of these works among the cited literature, they do not consider such methods as baselines (e.g., Hébert-Johnson, U., Kim, M., Reingold, O., & Rothblum, G. (2018, July). Multicalibration: Calibration for the (computationally-identifiable) masses. In International Conference on Machine Learning (pp. 1939-1948). PMLR., or Deng, Z., Dwork, C., & Zhang, L. (2023). Happymap: A generalized multi-calibration method. arXiv preprint arXiv:2303.04379.)

(3)- The authors evaluated the performance of the proposed method against the baselines based on three metrics. We would appreciate it if the authors would describe what aspects need to be assessed to guarantee the quality of the calibration and what metrics best assess those aspects. Furthermore, we would appreciate some insights into commonly used metrics to assess the general calibration and multicalibration scenarios so that they can be used as a reference when comparing to other works from this domain.

(4)- The authors did not perform any assessment on whether the results obtained are statistically significantly better than the baseline approaches. We suggest they perform such assessment.

**Other Comments Or Suggestions:**

(1) - What do the bolded results mean in the results tables? Why do we bolden results just for certain metrics? Please indicate with an arrow at the header, whether higher/lower results are better.

(2) - Table 2: align values to the right, so that differences in magnitude become evident.

(3) - Figure 1: we encourage the authors to provide the prediction bands of additional methods in order to enable a visual comparison between them and understand, in perspective, the goodness of the proposed method.

**Other Strengths And Weaknesses:**

We consider the paper to be clearly structured and address a relevant problem: providing Venn multicalibration guarantees. The authors made a great effort in providing theoretical demonstrations for their claims and to show how the Venn calibration relates to conformal prediction in multicalibration settings. Nevertheless, the experimental part of the paper is weak on the multicalibration aspect - one of the main contributions of the paper - an should be improved. Furthermore, the related work is weak considering that the main contribution refers to a novel method for multicalibration, but not many works and state-of-the-art from that domain are described.

**Questions For Authors:**

(1)- While the authors consider several datasets that have been used in reputed works related to conformal predictions, we would appreciate it if they could ground their choice from the subpopulations perspective. The fact that the datasets present subpopulations is key to their experiments for multicalibration and datasets failing to satisfy that characteristic should be excluded from the multicalibration experiments.

(2)- We would appreciate it if the authors would justify their choice regarding baseline methods, to understand why these are relevant to this particular experiments. In particular, we miss comparisons against multicalibration techniques. While the authors reference some of these works among the cited literature, they do not consider such methods as baselines (e.g., Hébert-Johnson, U., Kim, M., Reingold, O., & Rothblum, G. (2018, July). Multicalibration: Calibration for the (computationally-identifiable) masses. In International Conference on Machine Learning (pp. 1939-1948). PMLR., or Deng, Z., Dwork, C., & Zhang, L. (2023). Happymap: A generalized multi-calibration method. arXiv preprint arXiv:2303.04379.)

(3)- The authors evaluated the performance of the proposed method against the baselines based on three metrics. We would appreciate it if the authors would describe what aspects need to be assessed to guarantee the quality of the calibration and what metrics best assess those aspects. Furthermore, we would appreciate some insights into commonly used metrics to assess the general calibration and multicalibration scenarios so that they can be used as a reference when comparing to other works from this domain.

(4)- The authors did not perform any assessment on whether the results obtained are statistically significantly better than the baseline approaches. We suggest they perform such assessment.

(5)- While multicalibration requires a calibrator to be calibrated across subpopulations, the authors fail to identify subpopulations in each of the proposed datasets. We consider this to be a critical issue to provide insights regarding the quality of calibration for each of those subpopulations.

(6)- The related work mostly focuses on Venn and Venn-Abers calibration and their relationship to conformal predition, while the main contribution of the paper is Venn multicalibration. Therefore, we consider that the related work must be enhanced to introduce the reader to the state-of-the-art of the multicalibration techniques, showing the research gaps that exist in that domain and how the proposed method bridges them.

**Relation To Broader Scientific Literature:**

The authors provide many references to the relevant literature. Nevertheless, while they mostly focus on Venn and Venn-Abers calibration and their relationship to conformal predition, while one of their main contributions is Venn multicalibration. Therefore, we consider that the related work must be enhanced to introduce the reader to the state-of-the-art of the multicalibration techniques, showing the research gaps that exist in that domain and how the proposed method bridges them. Furthermore, the authors may be interested considering the following work: Haghtalab, N., Jordan, M., & Zhao, E. (2023). A unifying perspective on multi-calibration: Game dynamics for multi-objective learning. Advances in Neural Information Processing Systems, 36, 72464-72506.

**Theoretical Claims:**

(1)- The authors introduce a unified framework for Venn and Venn-Abers calibration, generalizing Vovk and Petej (2012) to arbitrary prediction tasks and loss functions. -> The authors demonstrate that their unified framework is a generalization of Venn and Venn-Abers calibration theoretically.

(2)- For quantile regression, the authors claim that Venn calibration corresponds to a novel CP procedure with quantile-conditional coverage, and that multicalibrated conformal prediction (Gibbs et al., 2023) is a special case of Venn multicalibration, unifying and extending existing calibration methods. -> The authors demonstrate their claim theoretically.

---

> ### Author Rebuttal · Authors · 2025-04-01
>
> Thank you very much for your detailed comments and suggestions. We will incorporate them in the revised version of the paper.
>
> We note our primary contribution is the generalization of Venn and Venn–Abers calibration to arbitrary loss functions. As a secondary contribution, we show that the same techniques yield a multicalibration algorithm with set-valued predictions. Our goal is primarily theoretical: to present a unified framework that generalizes and connects existing methods, including Venn–Abers calibration and conformal prediction.
>
>
>  1) In the revised paper, we will also report multicalibration error over specific subpopulations. In our experiments, we use the $\ell^2$ norm defined in Equation (4), corresponding to regression multicalibration error. This metric, also used in [1] for quantile multicalibration, ensures multicalibration over finite-dimensional classes of covariate distribution shifts—specific subpopulations being a special case. We define the function class in Equation (4) as linear combinations of binary features, one-hot encoded categorical features, and spline-transformed continuous features (with 5 knots). The reported $\ell^2$ error is the norm of the vector of calibration errors across these transformed features. For datasets with binary features, this directly captures multicalibration error over the associated subpopulations.
>
>
>
>
> 2)  We recognize that the evaluation of our multicalibration experiments can be improved, and we will address this in the revision. To the best of our knowledge, our Venn multicalibration algorithm is the only method that produces prediction sets capturing uncertainty under multicalibration in regression settings, and thus lacks direct baselines. In principle, our approach can be combined with any multicalibrator that outputs empirically multicalibrated point predictions—such as those in the cited references—to generate uncertainty-aware prediction sets. One goal of our experiments is to show how prediction set widths vary across datasets with different sample sizes and feature dimensions. For instance, the \textsc{Comm} and \textsc{Star} datasets yield wider prediction sets due to high feature dimensionality and small sample sizes, reflecting greater uncertainty in the multicalibration process.
>
>
>
> 3) Thank you for this suggestion. We agree that a clear discussion of calibration quality and appropriate evaluation metrics is important for interpreting and comparing results, and we will include this in the revised version. In the multicalibration experiments, our primary goal is to assess the quality of set-valued predictions in terms of (i) their size and (ii) the calibration error of the oracle multicalibrated prediction guaranteed to lie within the set. Specifically, we aim for multicalibration uniformly over target populations whose density ratios with the source population lie in a linear class derived from a basis transformation of the features. The reported $\ell^2$ multicalibration error measures how well the predictions satisfy the multicalibration criterion across many overlapping subgroups defined by these density ratios. This approach is motivated by prior work in quantile multicalibration [1] for feature-conditional coverage. We will incorporate this clarification in the final version to better guide practitioners and researchers seeking meaningful evaluation metrics for multicalibrated prediction sets.
>
>
> 4) Since the datasets are based on real-world data, we cannot generate independent replicates to compute standard Monte Carlo error estimates. Instead, we averaged all metrics over 200 random train-test-evaluation splits. In the final version, we will report Monte Carlo error estimates reflecting the variation of these metrics across the random splits.
>
> 5) In the revised paper, we will also report subpopulation multicalibration errors. The multicalibration definition in Equation (5) extends beyond fixed subgroups to classes of "covariate shifts", following the terminology of [1] on quantile multicalibration. Consistent with this work, we adopt a model- and subgroup-agnostic approach, aiming to approximate multicalibration across many subgroups without requiring explicit specification. In our experiments, we achieve this by calibrating over subgroups whose density ratios with the source population are additive functions of the covariates, implemented via adjustments over additive spline basis functions. While we agree that subgroup-specific calibration is also valuable, our experiments focused on guarantees that do not depend on manual subgroup definition.
>
>
> (6) In the revised version, we will add background on multicalibration and clarify how our contributions relate to prior work. While multicalibration is an important aspect, we note our primary focus is on calibration.
>
>
> [1] Gibbs, Isaac, John J. Cherian, and Emmanuel J. Candès. "Conformal prediction with conditional guarantees." arXiv preprint arXiv:2305.12616 (2023).

---

> > ### Comment · Reviewer_BPmy · 2025-04-09
> >
> > We thank the authors for their responses. We have read them and decided to maintain our score.

---

### Official Review · Reviewer_vJic · 2025-03-18

**Overall Recommendation:** 3

**Summary:**

Conformal Prediction arrises as a techniques for turning a point predictor to a set predictor, allowing for a guarantee that the set contains the ground-truth target with high probability. Venn-Predictor operates similarly but output a set of probabilistic prediction with the guarantee that at least one of them is calibrated. The core contribution is that the paper extends Venn calibration (originally for multi-class classification) to arbitrary prediction tasks and general loss functions. Provides a unified framework that applies to both classification and regression. Ensures finite-sample marginal calibration while allowing for asymptotic conditional calibration under stronger regularity .

**Claims And Evidence:**

Yes the assumptions and claims are transparently stated.

**Essential References Not Discussed:**

The recent advances on smooth ECE could be included but does not have negative consequence for understanding the paper.

**Experimental Designs Or Analyses:**

The evaluations does not provide errors bars, and compare the methods.

Plus, from table 1, the average width are quite all similar except in the Bike dataset.

The authors might want include several recents methods other than CQR (published at least 5 years ago). Overall the strict advantages of such proposed methods wrt classical CP is not that clear.

**Methods And Evaluation Criteria:**

The evaluation is reasonable but could be improved in several aspect i.e including more existing CP methods.

**Other Comments Or Suggestions:**

NA

**Other Strengths And Weaknesses:**

The overall proposition of extending VP with a specific loss is quite interesting on its own, but the authors could detail and motivate  a bit more this part.

The numerical experiments does not demonstrate a clear advantage of the proposed method but seems competitive.

**Questions For Authors:**

- I appreciate your work on generalized Venn and Venn-Abers calibration. I have some questions regarding the empirical calibration guarantee. From Condition C3, it appears that empirical calibration is enforced by exactly solving isotonic regression and recomputing it for each new test point X_{n+1}, as well as for every possible value of y. This suggests that calibration is dynamically maintained by including the test point in the calibration set before solving isotonic regression, avoiding the usual generalization issues. However, this approach raises computational concerns: (1) Since isotonic regression is recomputed exactly for each new test point, how does this scale in practice? (2) Given that recalibration is done for all possible values of y, does this lead to a significant computational burden, especially for large-scale settings? (3) Does this approach imply that we cannot precompute a single calibrated model but must instead recalibrate dynamically at inference time? I would appreciate any clarifications on these aspects. Overall I am confuse with the computational overhead, and do not see how techniques used in Full CP help Here.

- Often, one needs to have disjoint sets and not only a single interval. Can the proposed method allows such a flexibility?



"## update after rebuttal"

I maintain my score and previous points. I think the authors should clarify more the computational aspect, choice of the grid, how it impacts coverage, computational efficiency etc ... These issues are not fully solved for full conformal prediction, and a transparent discussions in the case of this particular paper could help readers.

**Relation To Broader Scientific Literature:**

Since the introduced method suffer from computational limitations as Full conformal prediction, discussing how to overcome the limitations could be welcomed.

Also, it is quite well known that calibration metric such as ECE suffers come inconsistency, it should be nice to include smooth version of ECE.

**Theoretical Claims:**

The claims and proofs essentially follows existing works on Venn-predictors with the inclusion of the loss function in the definition of calibration following (Whitehouse etal 2024). I followed the proof of the main results 3.1 which seems correct, and skip the rest.

The authors might want to revisit the claims of extending VP to *arbitrary loss function* since the latter is quite restricted (smooth, proper loss, finite moment assumptions, lipschitzness etc ...) etc...

---

> ### Author Rebuttal · Authors · 2025-04-01
>
> Thank you for the thoughtful questions and suggestions. We will take incorporate your suggestions in the revised version of our manuscript. We clarify the efficiency and practicality of our method below.
>
> 1. **Efficient Approximation and Scalability**
>    The algorithm can be efficiently approximated using the approach described in Section 3.4 of [1] for Venn-Abers calibration with squared error loss, which we also adopt in our implementation. The key observation is that the Venn-Abers algorithm only depends on the test point features through the original model prediction (which is crucially a one-dimensional quantity). As a result, we can discretize the one-dimensional prediction space—e.g., by binning into a grid of 200 bins—and run the algorithm once per grid point. Isotonic regression can be computed efficiently using regression trees with monotonicity constraints, such as those available in XGBoost. Across all datasets (each with tens of thousands of data points), computing the approximate Venn-calibrated sets for all points takes under a minute. This method scales well to large datasets via XGBoost. The isotonic regression step is computationally lightweight and can be parallelized if needed.
>
> 2. **Efficiency at Inference Time**   In our implementation, we precompute the Venn-Abers prediction sets over a discretized grid (200 bins) of model predictions. At inference time, we use nearest neighbor matching or linear interpolation to produce calibrated prediction sets, avoiding any calibration computation at run time. The approximation error from this approach is negligible in practice, likely due to the piecewise constant nature of both the isotonic regression solution and the Venn-Abers sets.
>
> 3. **Clarification on Disjointness**  We are unsure why disjointness would be required. The Venn-Abers prediction set is defined as the set of predictions obtained when calibrating with a new data point and all possible imputed outcomes. There is no requirement or expectation for this set to be disjoint.
>
> 4. **Improving experiments** We will add additional baselines to the conformal prediction experiments, including the conditional conformal prediction method of [1] and approaches based on alternative conformity scores, such as the normalized absolute residual error. We will also include error bars for our evaluation metrics to account for variability across the 200 train–validation–test splits over which the results are averaged.
>
>
>
> [1] van der Laan, Lars, and Ahmed M. Alaa. "Self-calibrating conformal prediction." arXiv preprint arXiv:2402.07307 (2024).
> [2] Gibbs, Isaac, John J. Cherian, and Emmanuel J. Candès. "Conformal prediction with conditional guarantees." arXiv preprint arXiv:2305.12616 (2023).

---

> > ### Comment · Reviewer_vJic · 2025-04-02
> >
> > Thanks for your comments.
> >
> > Having a disjoint set is not a requirement. However, one would like to have a union of confidence interval when the data is bimodal for example. I was wondering if your proposition allows such flexibility or it always output a single interval.

---

> > > ### Author Response · Authors · 2025-04-02
> > >
> > > Thank you for the clarification! This can indeed be done using the Venn/Venn–Abers procedure, and we will add a discussion of this in the revised version. The Venn–Abers procedure involves computing the calibrated prediction for all possible imputed values $y$ of the outcome $Y_{n+1}$ we wish to predict (lying in the space $\mathcal{Y}$). If the outcome space $\mathcal{Y}$ is a disjoint union of sets, then the Venn–Abers prediction set will also be a disjoint union of sets. This is a property also shared by conformal prediction. Our theoretical results only require that $\mathcal{Y}$ contains the true outcome $Y_{n+1}$, so the user is free to choose any such set. If the structure of the outcome space is not known a priori, a natural approach is to impute using the observed outcomes in $(Y_1, \dots, Y_n)$, which, for sufficiently large samples, will nearly contain $Y_{n+1}$ up to negligible discretization error.

---

### Decision · Program_Chairs · 2025-05-01

**Decision:**

Accept (poster)

**Comment:**

The submission extends Venn calibration from the multi-class setting to arbitrary prediction tasks, outputting a set of probabilistic predictors with the guarantee that at least one of them is calibrated.  This is an important topic, related to conformal prediction.  The claimed contribution is well placed, and supported primarily by theoretical analysis and a few empirical studies.  The reviewers were unanimous in their opinion that the submission be accepted.